# Efficient bubble/precipitate traffic enables stable seawater reduction electrocatalysis at industrial-level current densities

Jie Liang[1,2,6], Zhengwei Cai[1,6], Zixiao Li[2], Yongchao Yao ●[2], Yongsong Luo[2], Shengjun Sun[1], Dongdong Zheng[1], Qian Liu[3], Xuping Sun ●[1,2,4] ✉ & Bo Tang ●[1,5] ✉

Seawater electroreduction is attractive for future $H_2$ production and intermittent energy storage, which has been hindered by aggressive $Mg^{2+}/Ca^{2+}$ precipitation at cathodes and consequent poor stability. Here we present a vital microscopic bubble/precipitate traffic system (MBPTS) by constructing honeycomb-type 3D cathodes for robust anti-precipitation seawater reduction (SR), which massively/uniformly release small-sized $H_2$ bubbles to almost every corner of the cathode to repel $Mg^{2+}/Ca^{2+}$ precipitates without a break. Noticeably, the optimal cathode with built-in MBPTS not only enables state-of-the-art alkaline SR performance (1000-h stable operation at $-1\,A\,cm^{-2}$) but also is highly specialized in catalytically splitting natural seawater into $H_2$ with the greatest anti-precipitation ability. Low precipitation amounts after prolonged tests under large current densities reflect genuine efficacy by our MBPTS. Additionally, a flow-type electrolyzer based on our optimal cathode stably functions at industrially-relevant $500\,mA\,cm^{-2}$ for 150 h in natural seawater while unwaveringly sustaining near-100% $H_2$ Faradic efficiency. Note that the estimated price (~1.8 US\$/kg$_{H2}$) is even cheaper than the US Department of Energy's goal price (2 US\$/kg$_{H2}$).

$H_2$ is the cleanest energy carrier and a key building block in the manufacture of various essential chemicals, such as $NH_3$, $CH_3OH$, and petroleum products, but the traditional energy- and $CO_2$-intensive $H_2$-making processes with a limited $H_2$ product purity in fossil-fuel-based refineries are still in service today[1–3]. Water electrolysis, an appealing technolgy to extract high-purity $H_2$ using water and renewably generated electricity, is now expected to help the global need for decarbonization (i.e., hydrocarbon economy to hydrogen economy)[4–7], but the severe shortage and extremely uneven distribution of pure water resources contradict the worldwide deployment. Being the most abundant resource on Earth, seawater is widely accepted as

a promising alternative to pure water for feeding the water electrolyzers[8–13]. Increasing studies acknowledge that the electrosynthesis of $H_2$ gas in distributed coastal/offshore seawater splitting plants can be a highly desirable and practical solution in the future[14–24]. The electrified process of turning seawater to $H_2$, however, faces two significant obstacles: (1) cathodic precipitation of $Mg^{2+}/Ca^{2+}$ as well as (2) anodic corrosion induced by halides ($Cl^-$ and $Br^-$) and their derivates (e.g., $ClO^-$). Great recent attention has focused on designing anti-corrosion anodes that can yield $O_2$ from natural/alkaline seawater, including $Cr_2O_3–CoO_x$/Ti fibre felt[25], $Co_{3-x}Pd_xO_4$/Ni foam (NF)[26], glassy carbon/$MO_x$/CoPi (M = Pb, Mn, Fe, Cu)[27], NiIr-LDH/NF[28], RuMoNi/NF[29],

[1]College of Chemistry Chemical Engineering and Materials Science, Shandong Normal University, Jinan 250014 Shandong, China. [2]Institute of Fundamental and Frontier Sciences, University of Electronic Science and Technology of China, Chengdu 610054 Sichuan, China. [3]Institute for Advanced Study, Chengdu University, Chengdu 610106 Sichuan, China. [4]High Altitude Medical Center, West China Hospital, Sichuan University, Chengdu 610041 Sichuan, China. [5]Laoshan Laboratory, Qingdao 266237 Shandong, China. [6]These authors contributed equally: Jie Liang, Zhengwei Cai. ✉e-mail: xpsun@uestc.edu.cn; tangb@sdnu.edu.cn

reconstructed Ni-Fe oxyhydroxide/carbon cloth (CC)[30], NiFe oxalates/NF[31], NiFe-CuCo layered double hydroxide (LDH)/NF[16], $Ni_xCr_yO$/carbon paper (CP)[32], etc. While the performance of reported seawater oxidation anodes continues to be improved, the equally vital cathode performance, especially in natural seawater[33–42], is still at a subpar level with low geometric current density ($j$) and/or unsatisfactory stability. To go further, considering that the ultimate goal of seawater splitting is the electrosynthesis of $H_2$, the identification of genuine cathodes with sufficiently high activity and stability, particularly under realistic seawater electrolysis condition, is crucial. However, there have been fewer breakthroughs in the past few years regarding the cathode performance, especially the anti-precipitation ability, of electrochemical natural seawater reduction (eNSR).

Up to now, few cathodes can drive eNSR at high efficiency, like affording a $j$ of −500 mA cm$^{-2}$ with ~100% Faradic efficiency (FE) of $H_2$ during prolonged testing due to aggressive precipitation of $Mg^{2+}$/$Ca^{2+}$. In situ precipitation at the cathode occurs readily to block/poison catalytic sites when pH ≥ ~9.5, leading to voltage loss and inferior $H_2$-evolution FE[25,43]. As such, different research groups frequently utilize alkaline seawater to make $H_2$[44–52]. While electrochemical alkaline seawater reduction (eASR) induces less precipitation interference, eASR requires additional KOH and induces equipment corrosion. Moreover, alkaline seawater still contains residual $Mg^{2+}$/$Ca^{2+}$, which can build up over time as eASR-based overall water splitting still consumes water. More recently, confining $Mg^{2+}$/$Ca^{2+}$ or $OH^-$ to catalyst surfaces was reported to empower cathode anti-precipitation capability[25,53]. Yet, the tests were either performed in alkaline seawater or under a modest $j$ (i.e., 200 mA cm$^{-2}$). Additionally, another bottleneck impeding eNSR operation stems from the scarcity of buffering ions in natural seawater, which results in considerable energy consumptions due to high overpotentials ($\eta$). Therefore, the critical challenges in one-step eNSR to $H_2$ under an industrial-level $j$ do remain in the scientific designs of anti-precipitation and high-active cathodes.

In this work, we report an architecture with a microscopic bubble/precipitate traffic system (MBPTS) that proficiently integrates high-active catalytic sites from a classical $H_2$ evolution material, NiCoP, with a honeycomb-like porous carbon framework rich in non-random open micro-channels towards record-high eASR/eNSR performances. Specifically, massive small-sized $H_2$ bubbles are able to detach themselves from evenly distributed nanostructured NiCoP throughout the carbon frame material at the most appropriate migration velocity during seawater reduction, and unceasingly flow along channels/pores of the electrode-like cars on highways, forming strong localized airflows with homogenized forces that not only repel nearby Mg/Ca precipitates effectively but also vacate reaction sites timely. Any precipitates close to our cathode can be removed readily as bubbles are homogeneous in size and distribution and are modest in velocity. As a result, in alkaline seawater, our cathode attains an ampere-level $j$ of −1 A cm$^{-2}$ at a record low $\eta$ of only 160 mV and operates stably for at least 1000 h under −1 A cm$^{-2}$ with a negligible voltage loss. A flow-type electrolyzer catalyzed by our NiCoP-based 3D cathode further supports stable seawater electrolysis at 500 mA cm$^{-2}$ for 150 h, with symmetrical feeds of unprocessed natural seawater. Importantly, the electrode surface has less precipitates after the prolonged eASR and eNSR, thereby validating the superb self-cleaning ability from our MBPTS.

## Results

### A representative 3D integrated honeycomb-like architecture

Pinewood-derived carbon (PC) was synthesized by facile pyrolysis of treated pinewood. A digital image, scanning electron microscopy (SEM) images (Supplementary Figs. 1 and 2), pore structure data (Supplementary Fig. 3 and Table 1), and Fourier transform infrared (FTIR) (Supplementary Fig. 4) of PC reflect its properties including lightweight, free-standing, highly porous, and rich in functional groups, etc. A CoNi precursor (Supplementary Fig. 5) with nanoneedle

array morphological features was in situ hydrothermally created on the PC, which was subjected to a topotactic phosphidation step to obtain binder-free NiCoP decorated PC (denoted as NCP/PC). Besides, similar procedures were adopted to prepare monometallic phosphide samples, i.e., CoP/PC and $Ni_2P$/PC (Supplementary Fig. 6), as the contrast cathodes. The reasons for introducing such metal phosphides as catalytic materials on PC for subsequent eASR/eNSR experimental demonstrations are briefly described in Supplementary Note 1. The comprehensive structure/composition characterizations of the representative NCP/PC were performed. The crystal nature of NCP/PC was studied by X-ray diffraction (XRD) analysis (Fig. 1a), and four notable reflection peaks at 40.9°, 44.8°, 47.4°, and 54.2° belonging to (111), (201), (210), (300) planes of hexagonal NCP (JCPDS 001-5135) can be observed, except for carbon peaks at ~23° and ~40° from the PC. For the carbonized skeleton of PC, its top surface exhibits a high density of clearly defined, aligned micro-channels, with a clean and relatively flat surface (Figs. 1b–d). Growth of homogeneously/densely distributed nanostructured NCP on PC in both parallel and vertical directions of the channels is visible in the SEM images (Figs. 1e–g) without clogging open micro-channel systems of PC that resemble honeycombs. Consequently, walls of the honeycomb skeleton with NCP (Fig. 1f) do thicken by about a few micrometers, suggesting ~1.95 μm in an average length of the NCP. Moreover, PC alone shows an unsmooth and bare vertical cutting surfaces with shallow and elongated grooves (Fig. 1h, i), whereas a substantial number of needle-like NCP nanostructures are present on the vertical cutting surfaces of the NCP/PC (Fig. 1j, k). Thus, uniformly and densely distributed NCP as reaction sites nicely encircles the entire carbon skeleton, according to SEM characterizations for NCP/PC. Transmission electron microscopy (TEM) image once more supports the nanoneedle-like structures of NCP (Supplementary Fig. 7), and the related high-resolution TEM (HRTEM) images display lattice spacing of 0.262 nm (Fig. 1l) and 0.232 nm (Fig. 1m), which is specific to (210) and (111) planes of NCP, respectively. Moreover, SEM and energy-dispersive X-ray (EDX) mapping of NCP/PC (Fig. 1n) show the even distributions of Ni, Co, P, O, and C across the entire open channels, implying an oxidized surfaces and highly dispersed NCP on PC. Note that the NCP/PC also contains trace elemental N from the PC (Supplementary Fig. 8). The surface chemical states for NCP/PC were unveiled by X-ray photoelectron spectroscopy (XPS, see wide energy range data in Supplementary Fig. 9). Deconvoluted Ni $2p_{3/2}$ peaks (Fig. 1o) imply the presence of partially charged $Ni^{\delta+}$ species as well as Ni-oxidized states ($Ni^{3+}$ and $Ni^{2+}$)[54–56]. Similarly, deconvoluted Co $2p_{3/2}$ peaks (Fig. 1p) are assignable to $Co^{\delta+}$ in NiCoP and oxidized Co species having positive states ($Co^{3+}$ and $Co^{2+}$)[57–59]. Superficial oxidation of the NCP/PC results in the strong signal of phosphate species (i.e., $P^{5+}$, P − O) and the weak signals of P $2p_{3/2}$ and P $2p_{3/2}$ (i.e., $P^{\delta-}$, Co−P, Ni−P) in the P $2p$ spectrum (Fig. 1q)[60]. Such observation is in line with the two dominant peaks occurring in the O $1s$ spectrum (Fig. 1r)[61,62], which may represent oxidized phosphate species (O1) and surface-adsorbed oxygen species (O2). The mechanical properties of NCP/PC and PC were further evaluated using three-point bending tests, with strains at failure in the 1.37-to-1.6% range, a higher flexural strength value for PC and a higher flexural modulus for NCP/PC (Supplementary Fig. 10). Besides, a piece of NCP/PC can bear the strain of a 200-g weight pulling on it (Supplementary Fig. 11). Furthermore, NCP/PC has a higher conductivity than bare PC (0.03153 KS cm$^{-1}$ versus 0.0019 KS cm$^{-1}$), according to data of the four-probe conductivity tests (Supplementary Fig. 12).

### Outstanding eASR performance

$H_2$-evolution activities of the bare PC, PC-loaded commercial Pt/C (Pt/C/PC), NCP/PC and the monometallic counterparts (i.e., CoP/PC and $Ni_2P$/PC) were evaluated in a 1 M KOH + seawater electrolyte over the range of 0 ~ −0.6 V versus a reversible hydrogen electrode ($V_{RHE}$, Fig. 2a). All recorded currents were normalized by the cathode

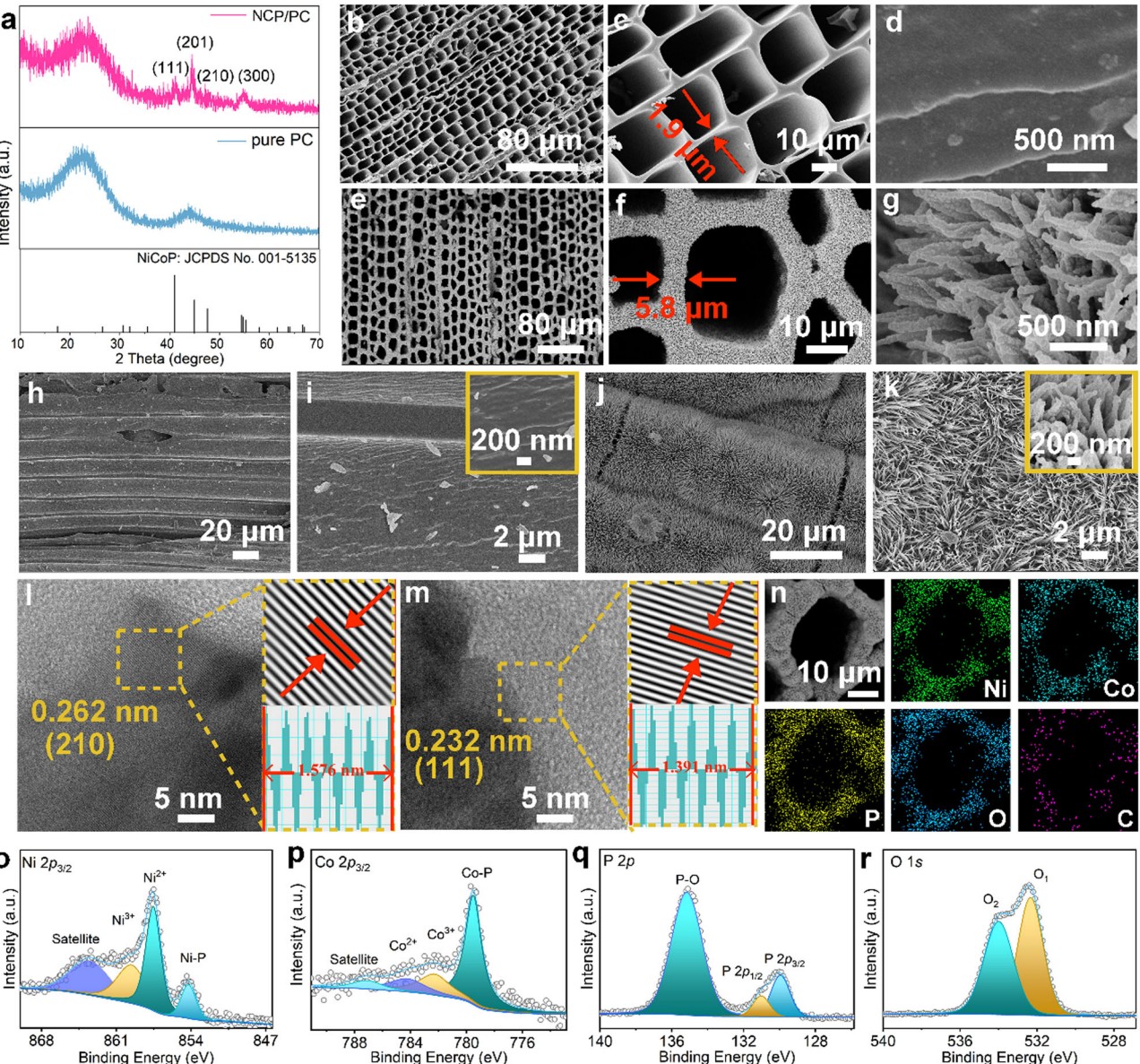

**Fig. 1 | Structure characterizations for honeycomb-like NCP/PC and PC. a** XRD patterns for NCP/PC and PC. **b–d** Top view of PC. **e–g** Top view of NCP/PC. **h, i** Cross-section view of PC. **j, k** Cross-section view of NCP/PC. **l, m**)HRTEM images with the corresponding line scan intensity profiles. **n** SEM and the related elemental mapping images. Ex situ XPS spectra of NCP/PC in the (**o**) Ni $2p_{3/2}$, (**p**) Co $2p_{3/2}$, (**q**) P $2p$, and (**r**) O $1s$ regions.

geometric areas. The thickness of our NCP/PC is even lower than the commercial Ni foam (Supplementary Fig. 13). PC itself with an inferior eASR activity only delivers low $j$, not reaching −100 mA cm$^{-2}$ even at −0.6 $V_{RHE}$. All three phosphide-based electrodes exhibit comparable activities to the Pt/C/PC and are more effective than Pt/C/PC at attaining high $j$ due to the dramatic rise in the eASR current (Fig. 2a). Voltammetry curves for the three phosphide-based cathodes fluctuate due to their violent release of gaseous H$_2$. NCP/PC requires lower $\eta$ (92 mV, 121 mV, and 145 mV) to reach the $j$ of −100 mA cm$^{-2}$, −500 mA cm$^{-2}$, and −1000 mA cm$^{-2}$ compared to CoP/PC (138 mV, 218 mV, and 270 mV) and Ni$_2$P/PC (202 mV, 295 mV, and 330 mV, Fig. 2b). The least tendency for $\eta$ to rise with the $j$ is realized by NCP/PC, thereby reflecting the extraordinary capacity to boost $j$ markedly while with a bit of voltage input. In comparison to CoP/PC and Ni$_2$P/PC, the NCP/PC exhibits the highest double-layer capacitance ($C_{dl}$, 91.81 mF cm$^{-2}$), nearly 1.37 and 2.95 times greater (Supplementary Fig. 14). Electrochemical impedance spectroscopy (EIS) data indicate the superior charge transfer kinetics for NCP/PC compared to the two

counterparts as well (Supplementary Fig. 15). NCP/PC has a sharply decreased Tafel slope of 40.93 mV dec$^{-1}$ compared to those of CoP/PC (104.81 mV dec$^{-1}$) and Ni$_2$P/PC (124.53 mV dec$^{-1}$) under the eASR conditions (Fig. 2c). This value is rather close to theoretical ~40 mV dec$^{-1}$ at which a charge-transfer Heyrovsky process, i.e., adsorbed atomic H (H$_{ads}$) + H$_2$O + e$^-$ → * + H$_2$ + OH$^-$, turns to the rate-determining step (RDS)[63]. Such a Tafel slope drop thus states that the Volmer reaction (i.e., the cleaving of the H − OH bond into H$_{ads}$ and OH$^-$) on our NCP/PC is no longer a significant barrier for the overall H$_2$ evolution chemistry. Therefore, we highlight that the Tafel slope recorded on the NCP/PC in alkaline seawater (40.93 mV dec$^{-1}$) not only exceeds by far the Tafel slopes of many previously reported eASR cathodes (Supplementary Table 2) but also is considerably smaller than those of the reported NiCoP-based electrocatalysts like NiCoP/Ti$_3$C$_2$T$_x$ Mxene (77.3 mV dec$^{-1}$) and NiCoP/C nanobox (96 mV dec$^{-1}$) in pure water-based 1 M KOH solution[58,64]. This low Tafel slope confirms that even under challenging eASR conditions, the NCP/PC inherently affords a more excellent H$_{ads}$ coverage ($\theta_H$) for H$_2$ production, which is likely ascribed

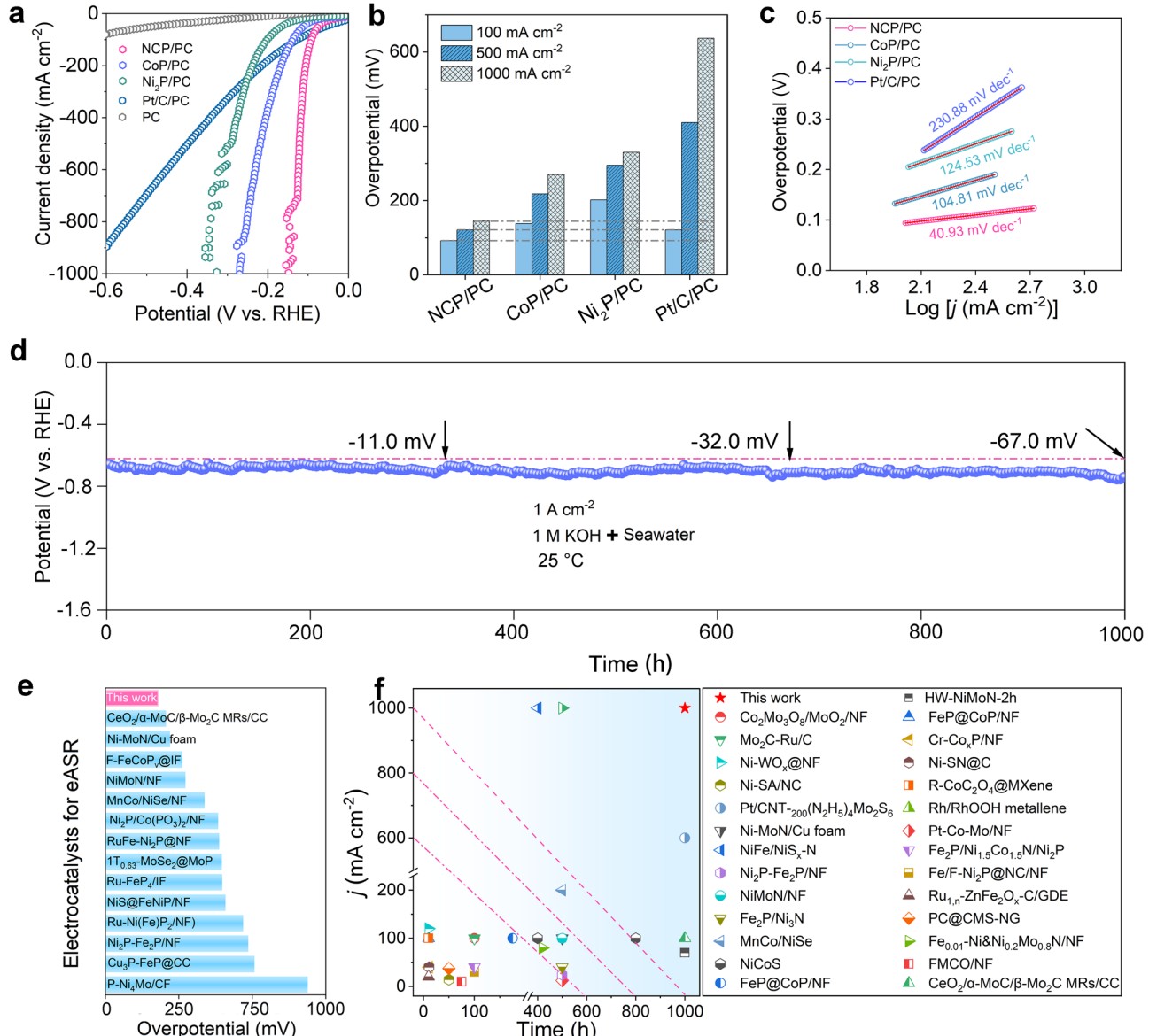

**Fig. 2 | eASR performance. a** Linear sweep voltammetry curves of NCP/PC, Ni₂P/PC, CoP/PC, Pt/C/PC, and bare PC. **b** Comparisons of $\eta_{100}$, $\eta_{500}$, and $\eta_{1000}$ for different cathodes. **c** Tafel slopes obtained from polarization curves. **d** Chronopotentiometry curve of NCP/PC towards long-term eASR at the fixed $j$ of −1 A cm⁻². **e** Comparisons of $\eta_{500}$ for our NCP/PC with various state-of-the-art eASR cathodes. **f** Lifespan comparisons for our NCP/PC with various state-of-the-art cathodes in alkaline seawater. The catalyst mass loading for the optimal sample NCP/PC is -0.0042 g cm⁻². The pH of the electrolyte is -14. The charge transfer resistance of NCP/PC is 2.5 Ω.

to, but not limited to (1) far more accessible catalytic sites, (2) timely gas release (see later real-time optical microscopy results and water/gas contact angle data) on the porous 3D electrode geometrical structure, and (3) a possible nice electrical coupling of NCP to the underlying PC. As shown in Fig. 2d, our NCP/PC succeeds in demonstrating a 1000-h eASR process under the ampere-level $j$ of −1 A cm⁻². The most significant point is that the NCP/PC is far more stable than practically all previous eASR cathodes, with the voltage increases of only 11 mV after 330 h of operation and only 67.0 mV after 1000 h. We provided comparison images using the cutting-edge eASR cathodes to illustrate once more the exceptional eASR performance of our NCP/PC. Various advanced cathodes were designed for the eASR over the past few years, and cathodes like the NCP/PC with highly competitive activities in alkaline seawater has rarely been reported (Figs. 2e, f and Supplementary Table 3, 4). The $\eta$, particularly the $\eta$ to achieve ampere-level $j$, is a crucial parameter for saving electricity power in practical

electrolysis. Noticeably, a comparative histogram of $\eta_{1000}$ levels (Fig. 2e) suggests that the NCP/PC has substantially lower $\eta_{1000}$ than many other state-of-the-art cathodes in alkaline seawater, again confirming that NCP/PC is no doubt one of the most efficient one with superior eASR efficiency at ampere-level $j$. The outstanding eASR lifespans achieved by NCP/PC under ampere-level $j$ presents a great progress (Fig. 2f and Supplementary Table 3). We must point out that this is the best performance of metal phosphide-based cathodes for eASR. Therefore, our work is of great significance for the development of metal phosphides and their nanocomposites in electrocatalytically driving the seawater reduction[19,65].

**Comparative studies to showcase genuine self-cleaning capabilities in eNSR**

After establishing superior eASR electrolysis performances for NCP/PC, the eNSR activities were then investigated. Expectedly, the activity

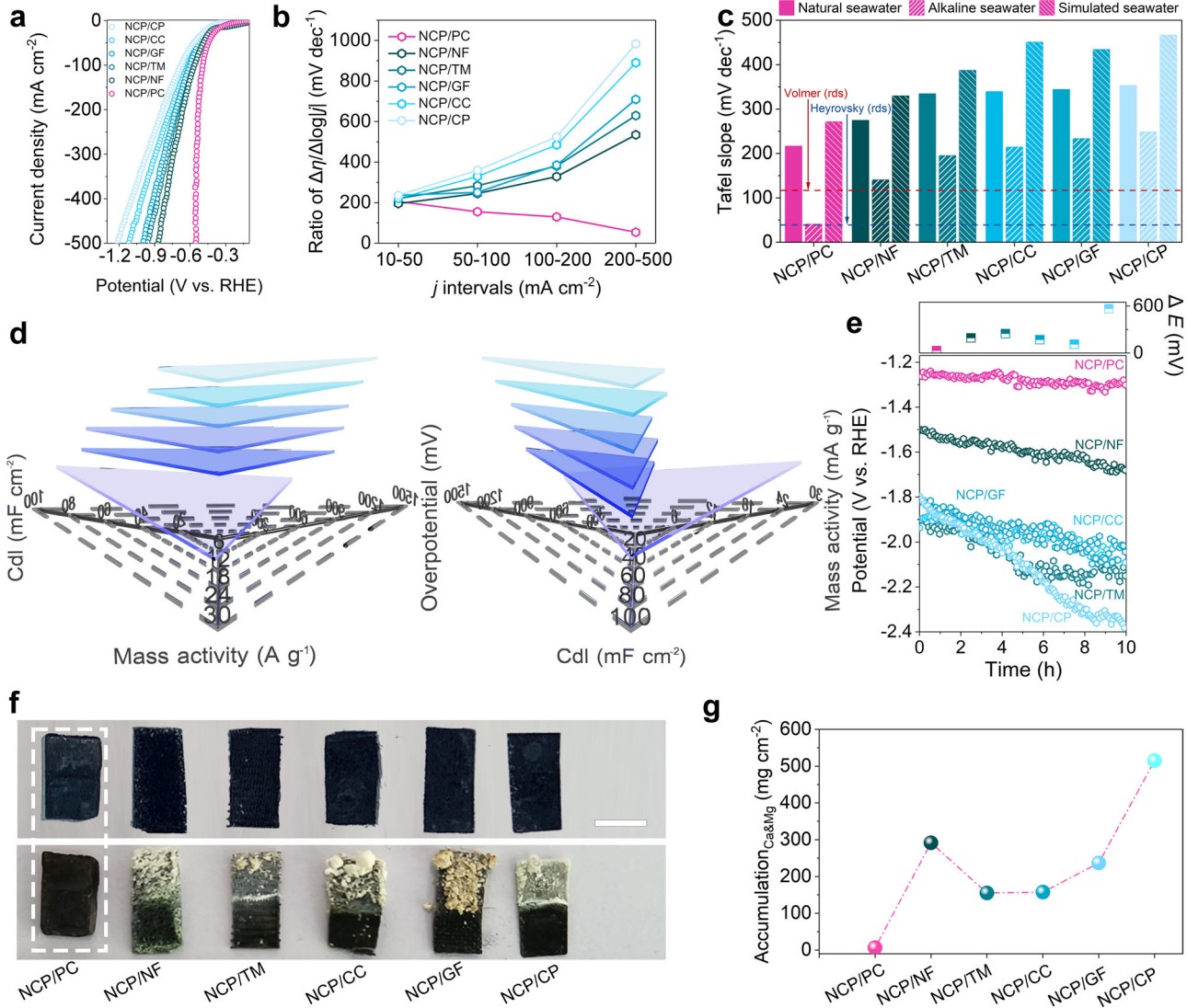

**Fig. 3 | eNSR performance comparison of NCP on various substrates. a** Ambient polarization curves (with *iR* correction) in natural seawter. **b** $\Delta\eta/\Delta log|j|$ ratios of NCP-based cathodes under a series of *j* intervals. **c** Tafel slopes in the varied situations. **d** Visual comparisons of $C_{dl}$, mass activity, and $\eta$ for various cathodes. **e** Chronopotentiometry curves for various cathodes and the corresponding $\Delta E$. **f** Pictures of various cathodes after the 10 h of eNSR. Scale bar: 0.5 cm. **g** $A_{Ca\&Mg}$ values for various cathodes after the long-term eNSR testing. For **f** from the left to the right are: NCP/PC, NCP/NF, NCP/TM, NCP/CC, NCP/GF, and NCP/CP.

of the NCP/PC in natural seawater still far exceeds those of other electrodes, including PC, Pt/C/PC, Ni₂P/PC, and CoP/PC (Supplementary Fig. 16). As such, we prepared a set of NCP-based electrodes to assess their performances in natural seawater comprehensively. Figure 3a shows the *E-j* curves of NCP supported by several kinds of substrates, including CP, CC, graphite felt (GF), Ti mesh (TM), NF, and PC towards eNSR. All other NCP-based cathodes were prepared using the same method as NCP/CP but with different substrates. The successful growth of nanoneedle-like NCP on CP, CC, GF, TM, and NF were verified with XRD and SEM (Supplementary Figs. 17 and 18) characterizations. As expected, NCP/PC outperforms all other five NCP-based samples in many metrics considered when evaluating eNSR performance (Fig. 3). While loaded with the same bimetallic phosphide, NCP/PC far excels all NCP-based counterparts, pushing the onset potential ($E_{onset}$) to more positive voltages and dramatically boosting the *j* (see the pink curve in Fig. 3a). In particular, the polarization curve of NCP/PC drops almost vertically around −0.3 V$_{RHE}$ to −0.6 V$_{RHE}$ and attains −500 mA cm⁻² at a low $\eta$ of only ~550 mV, indicating superb eNSR activity. Moreover, only the $\Delta\eta/\Delta log|j|$ ratio for NCP/PC decreases with the increase of the *j*, while $\Delta\eta/\Delta log|j|$ ratios for

five NCP-based counterparts increase significantly (Fig. 3b). This confirms the point that perks of eASR operation on NCP/PC under high *j* will be more apparent due to the good mass transfer ability, which is crucial for practical seawater electrolysis. As shown in Supplementary Fig. 19, geometric areas for NCP/PC are increased from 0.25 to 9 cm², yet this does not considerably lower the corresponding electrode activities. Moreover, the Tafel slopes of all NCP-based cathodes in different electrolytes (Fig. 3c), i.e., natural seawater, alkaline seawater, and simulated seawater (0.5 M NaCl) are contrastively analyzed in detail. Except for the NCP/PC and NCP/NF, all other NCP cathodes show Tafel slopes of around or above 200 mV dec⁻¹ in alkaline seawater, indicating their sluggish initial water dissociation step kinetics as well as higher energy barriers. In sharp contrast, only the Volmer limited mechanism (H₂O + e⁻ → OH⁻ + H$_{ads}$) achieved with the NCP/PC is far lower than 120 mV dec⁻¹ and approaching 40 mV dec⁻¹, thereby confirming more sufficient surface-localized proton supply for NCP/PC. Further, the Tafel slopes for all cathodes recorded in 0.5 M NaCl solution are inferior to those in natural seawater, probably stemming from the greater abundance of ions in natural seawater. In addition to optimizing elementary steps, developing NCP on PC can significantly

improve overall eNSR activities (Fig. 3d and Supplementary Table 5), including the highest mass activity (26.733 A mg$^{-1}$ at $-0.5$ V$_{RHE}$), lowest $\eta$ (553.8 mV at 500 mA cm$^{-2}$) and highest $C_{dl}$ (86.1 mF cm$^{-2}$). Such soaring overall activities are far superior to those of other NCP cathodes (e.g., the mass activity of 2.348 A mg$^{-1}$, $\eta$ of 992.6 mV at 500 mA cm$^{-2}$, $C_{dl}$ of 55.2 mF cm$^{-2}$ for the NCP/GF), showing a nice synergy catalysis effect between the NCP and PC. Notably, the $C_{dl}$ value of NCP/PC is 1.19 times as high as NCP/NF, 1.25 times as high as NCP/TM, 1.56 times as high as NCP/GF, 2.42 times as high as NCP/CC, and 2.83 times as high as NCP/CP (Supplementary Figs. 20 and 21). The highest $C_{dl}$ value reflects that the construction of the 3D NCP/PC architecture supplies more catalytically active sites for the adsorption/dissociation of H$_2$O and the further conversion of intermediates in the eNSR process, which is in line with the lowest Tafel slope achieved by NCP/PC. Moreover, NCP/PC exhibits the highest electrochemical active surface area (ECSA)-fitted $j$ as well (Supplementary Fig. 22). The charge-transfer resistance ($R_{ct}$) was determined to be 17.3, 16.8, 15.5, 14.2, 12.2, and 9.0 $\Omega$ for NCP/CP, NCP/GF, NCP/CC, NCP/TM, NCP/NF, and NCP/PC, separately, suggesting the facilitated charge-transfer process at the interface of NCP on PC (Supplementary Fig. 23 and Table 6). In addition, ample Mg$^{2+}$/Ca$^{2+}$ in seawater can interact with anions like OH$^-$ to create hydroxide precipitates that adhere to or even completely cover/obstruct the active sites, impairing the cathode's capacity to synthesize H$_2$. Significantly, during the 10-h eNSR electrolysis at the fixed $j$ of up to $-0.5$ A cm$^{-2}$, a low electrode potential ($E$) of ~$-1.25$ V$_{RHE}$ for our NCP/PC is well-maintained to give a reliable H$_2$ electrosynthesis process without evident voltage loss ($\Delta E$), whereas other electrodes exhibit striking $\Delta E$ to varying degrees (Fig. 3e). As displayed in Fig. 3f, only the surface of NCP/PC is clean as before testing after the long-period of electrolysis, while other five NCP-based cathodes are covered with thick and dense Mg$^{2+}$/Ca$^{2+}$ precipitates. We suggest a criterion, the accumulation amount of Mg and Ca ($A_{Ca\&Mg}$), which is obtained by dividing the Mg and Ca contents of the used cathode (mg) by the corresponding the electrode area (cm$^2$) in order to quantitatively assess the anti-precipitation ability and even indirectly reflect the impact of surface precipitation on performance. Noticeably, $A_{Ca\&Mg}$ values calculated from the inductively coupled plasma optical atomic emission spectrometer (ICP-AES) data (Fig. 3g) verify that only NCP/PC is immune from being substantially covered by Mg$^{2+}$/Ca$^{2+}$ precipitates (only ~6 mg cm$^2$), thereby confirming that the losses of activity for other NCP-based cathodes in the stability tests should be correlated with their high $A_{Ca\&Mg}$ values (~292 mg cm$^2$ for NCP/NF, ~155 mg cm$^2$ for NCP/TM, ~157 mg cm$^2$ for NCP/CP, ~267 mg cm$^2$ for NCP/GF, ~515 mg cm$^2$ for NCP/CC). Such surface deposits are mainly made of Mg(OH)$_2$, CaCO$_3$, and Ca(OH)$_2$ (see possible formation causes in Supplementary Note 2, for instance: OH$^-$ + HCO$_3^-$ → H$_2$O + CO$_3^{2-}$, Ca$^{2+}$ + CO$_3^{2-}$ → CaCO$_3$). XRD patterns for all NCP-based electrodes after 10-h eNSR (Supplementary Fig. 24) confirm that NCP/PC possesses exclusively high resistance to the Ca$^{2+}$/Mg$^{2+}$ precipitation, in contrast to other NCP-based cathodes. Notable diffraction peaks belonging to Ca$^{2+}$/Mg$^{2+}$ precipitates appear for the NCP/TM, NCP/GF, NCP/CC, NCP/CP, and NCP/NF, while the absence of peaks of precipitates in the XRD pattern for the NCP/PC indicates that only the NCP/PC possesses the strong anti-precipitation ability. Additionally, the elemental distribution of Mg and Ca for NCP/PC after the long-term eNSR confirms again the anti-precipitation capability of the NCP/PC, which show little Mg/Ca contents (Supplementary Fig. 25). We performed in situ Raman spectroscopic measurements on all NCP-based electrodes to characterize surface insoluble precipitates further. At the beginning of eNSR, Raman spectra of most electrodes show peaks at around 450 cm$^{-1}$, which should be attributed to phosphides (Supplementary Fig. 26)[56]. However, with the extension of electrolysis time, most NCP-based electrodes lose the peak signals of M − P, which implies that there is no metal phosphide on their surfaces. For instance, NCP/NF loses the M − P peaks after 30-min of electrolysis

(Supplementary Fig. 26b). By contrast, even though the peak intensities of M − P for NCP/PC decline, the peaks remain visible even after 60-min electrolysis, revealing the greatest anti-precipitation ability. The spectra of NCP/NF, NCP/CC, NCP/GF, and NCP/CP all show a clear peak at around 669 cm$^{-1}$, which may indicate the production and build-up of surface Ca(OH)$_2$ (Supplementary Fig. 26). Noticeably, except for NCP/PC, all NCP-based counterparts show visible Raman bands in between ~1000 cm$^{-1}$ and ~1300 cm$^{-1}$, further indicating that the surface precipitates should contain CaCO$_3$ and Ca(OH)$_2$ (Supplementary Fig. 26). In addition, Raman data in higher wavenumbers ranging from 2000 to 4000 cm$^{-1}$ reveal that NCP/NF, NCP/TM, NCP/CC, NCP/GF, and NCP/CP are all covered by insoluble Mg(OH)$_2$ (Supplementary Fig. 27). In sharp contrast, Raman spectra of NCP/PC does not show any peaks in this wavenumber range, fully indicating Mg(OH)$_2$-free surfaces. Except for time-dependent data in Supplementary Figs. 26 and 27, potential-dependent Raman analysis of NCP/PC reveals little surface precipitation as well (Supplementary Fig. 28). Afterwards, we measured the $C_{dl}$ of all NCP-based electrodes after the 10-h eNSR operation (Supplementary Figs. 29 and 30). Noticeably, changes in $C_{dl}$ value ($\Delta C_{dl}$) are 3.2 mF cm$^{-2}$ for NCP/PC, 67.3 mF cm$^{-2}$ for NCP/NF, 26.1 mF cm$^{-2}$ for NCP/CP, 33.8 mF cm$^{-2}$ for NCP/CC, 49.8 mF cm$^{-2}$ for NCP/GF, and 64.6 mF cm$^{-2}$ for NCP/TM. The NCP/PC shows the smallest $\Delta C_{dl}$, which is in line with minimal surface precipitation. Moreover, the polarization curves fitted by the ECSA data before and after the long-term testing basically overlap with each other (Supplementary Fig. 31), thereby indicating little decrease in the intrinsic activity of the catalytic reaction sites. Since all NCP-based counterparts share a similar nano-needle morphology (Supplementary Fig. 18) still perform poorer than NCP/PC, the all-round improvement in the activity, stability and precipitate-repelling ability for our NCP/PC reflects the critical function of the PC substrate.

## Effective traffic of bubble/precipitate towards superb eNSR performance

The key and most valuable product of seawater reduction is gaseous H$_2$, and how these bubbles behave and transport will directly and significantly affect how well the eNSR cathodes perform in the end, especially given that bubbles are more likely to obstruct the catalytically active sites under the conditions of industrial-level $j$. Moreover, the in situ formation of the Mg$^{2+}$/Ca$^{2+}$ precipitates nearby the cathode during the direct seawater reduction process would make the role of bubbles more pronounced, as bubbles with inherent buoyancy as well as additional forces from their coalescence/burst may act as the perfect nano/micro-sized helper for cleaning the cathode. However, there lacks comprehensive study on how to fully utilize gas release to mitigate precipitation actions on the seawater reduction cathode. Here, high-speed cameras were employed to capture real-time results of the release behaviors of bubbles from various cathodes and the transport of small precipitates that follow the gas movements. Monolithic electrodes were placed upright in natural seawater for electrolysis under the $j$ of 500 mA cm$^{-2}$. As shown in Fig. 4a–g, the NCP/PC surface releases the most uniformly sized and evenly dispersed bubbles at the highest efficiency (Fig. 4a), and the sizes of the majority of these bubbles do not significantly change as they rise (Supplementary Movie 1 and Supplementary Figs. 32 and 33). Since the bubbles are of uniform size and distribution, they are capable of pushing away any precipitate that is present around the NCP/PC effortlessly and promptly. Moreover, these massively moving bubbles successfully avoid the blockage of the active sites of NCP/PC, as evidenced by the superior current increments (Fig. 3a–c) as well as excellent stability (Fig. 3e). The remaining NCP cathodes (Fig. 4c–g), however, are incapable of cleaning themselves spontaneously, and their surfaces are covered with irregular and unevenly distributed bubbles. Specifically, NCP/NF (Fig. 4c) is basically unable to clean itself, which was exposed to heavy precipitates during the eNSR (Supplementary Movie 2 and

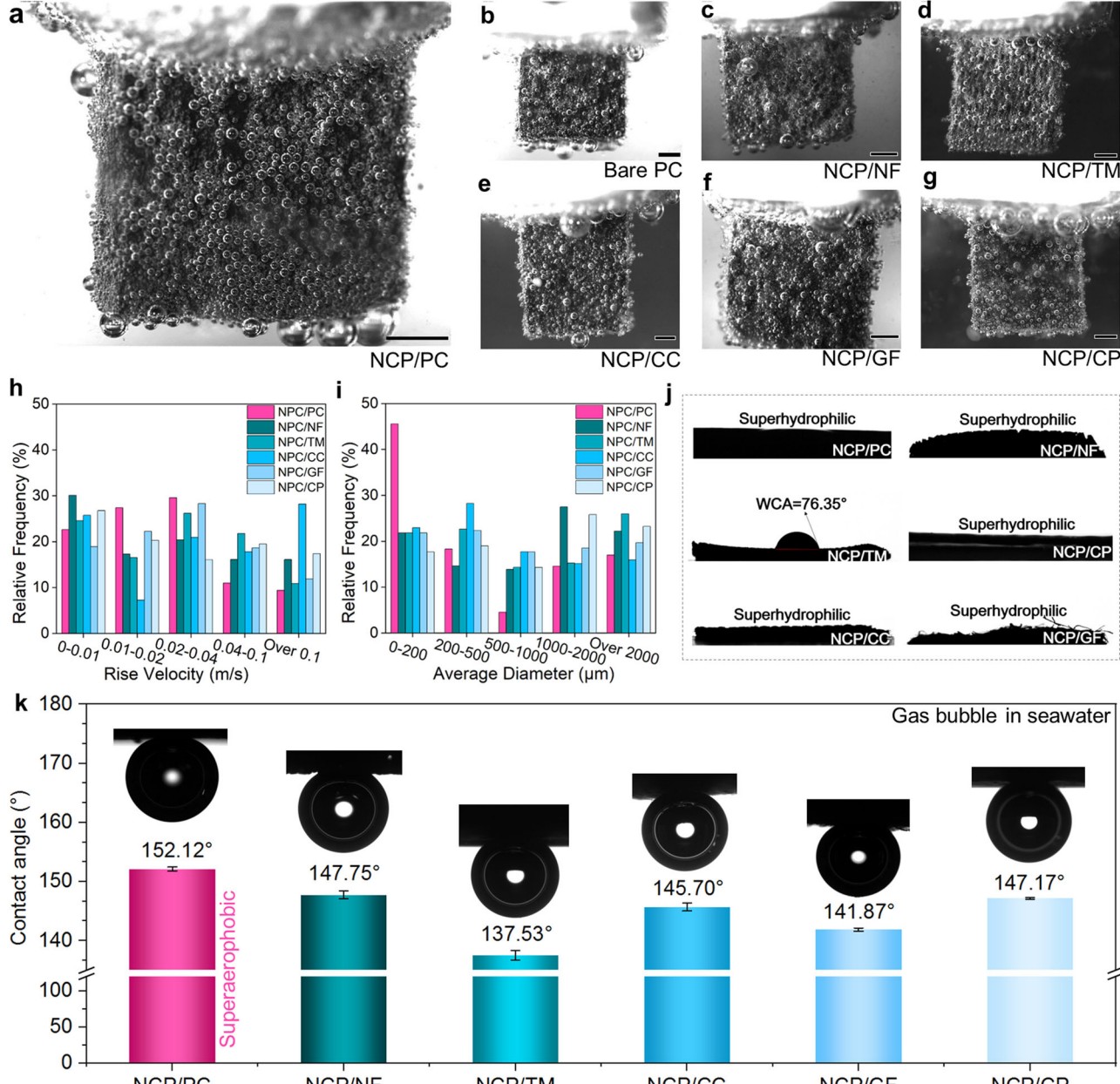

**Fig. 4 | Operando and ex situ investigations into the bubble behaviors of different electrodes. a–g** Gaseous $H_2$ release behaviors on NCP/CP, PC, NCP/NF, NCP/TM, NCP/CC, NCP/GF, and NCP/CP at the $j$ of −500 mA cm$^{-2}$ (from the front view). Due to the limitations of current observation technologies (we have already used one of the most advanced microscope technologies in this work to observe the moving bubbles), we can only observe micrometer-sized bubbles under the condition of industrial-level current density electrolysis. All the scale bars in (**a–g**) are 1 mm. **h** Statistical results of the rise velocities of $H_2$ bubbles for various eNSR cathodes. **i** Statistical results of the average $H_2$ bubble diameters for various eNSR cathodes. **j** Water contact angle data. **k** Underwater gas-bubble contact angle data. Error bars represent the standard deviation for triplicate independent measurements.

Supplementary Figs. 34 and 35). Also, huge bubbles released from NCP/NF remove a tiny portion of the precipitates, thus leaving a significant amount of precipitates to be continuously accumulated on its surfaces. NCP/TM (Fig. 4d) shows a strong adhesion to bubbles, and only large-size bubbles can move upwards quickly on it (Supplementary Movie 3 and Supplementary Figs. 36 and 37). Similarly, tiny bubbles emerge from the interior of the NCP/CC and then merge into larger bubbles before they begin to move upward (Supplementary Movie 4 and Supplementary Figs. 38 and 39). The white precipitates that constantly build up on the NCP/CC during the eNSR process indicate that it is not a self-cleaning cathode. Due to the uneven located and irregularly sized bubbles of NCP/GF (Supplementary Movie 5 and Supplementary Figs. 40 and 41), severe precipitation also occurs on it during the eNSR. For NCP/CP, there are considerable amounts of surface precipitates (Fig. 4g) in that sizeable surface parts are exposed to precipitates directly throughout the testing and fewer bubbles are released from its bottom (Supplementary Movie 6 and Supplementary Figs. 42 and 43). We further recorded and statistically analyzed the bubble release behaviors on all six NCP-based cathodes during continuous electrolysis. As show in Fig. 4h, excessive velocities tend to come from electrodes with poor anti-precipitation abilities, such as NCP/CC, while too slow-rise velocities come from problems with the structure of the electrodes themselves and the obstruction of the bubbles by precipitates or other forces (such as NCP/NF), with either too fast or too slow velocities being detrimental to the repulsion of the precipitates. Only the bubbles released by the NCP achieve the

most modest migration velocity. As show in Fig. 4i, only NCP/PC releases gas bubbles with the smallest average diameter, which should homogenize the force-repelling precipitates. The unique and superior ability of NCP/PC to repel $Mg^{2+}/Ca^{2+}$ precipitates among all NCP-based cathodes directly highlights the exclusive structural advantages of PC. As such, we investigated more PC-based electrodes. The first to be studied was the bare PC, which has a 3D porous structure similar to that of NCP/PC but much low eNSR activity. The upward migration of $H_2$ bubbles escaping from the surface of the bare PC (Fig. 4b) is so free that they appear to be unimpeded (Supplementary Fig. 44), allowing the precipitates near the PC to be cleaned off in time. At the bottom of PC, larger-sized bubbles are occasionally formed as well, which help to further push away precipitates that are slowly moving upward. It is clear that CP itself can provide certainly good anti-precipitation ability. Except for the bare CP electrode, we further tried other materials with electrochemical $H_2$ evolution activity, such as CoP, $Ni_2P$, and $MoS_2$, to integrate with the PC for eNSR. We prepared these materials onto CP as comparison samples to see whether constructing electrodes employing PC as the substrate naturally provides the structural benefit of repelling $Mg^{2+}/Ca^{2+}$ precipitates. Highly encouragingly, the CoP/PC, $Ni_2P/PC$, and $MoS_2/PC$ all achieve much better resistance to the $Mg^{2+}/Ca^{2+}$ precipitation compared to their CP-based counterparts, i.e., CoP/CP, $Ni_2P/CP$, and $MoS_2/CP$, suggesting that our PC-based MBPTS design does show a wide universal applicability (Supplementary Fig. 45). Moreover, we point out that despite the fact that CoP/PC, $Ni_2P/PC$, $MoS_2/PC$, and even bare PC all possess excellent anti-precipitation ability, NCP/PC outperforms them in terms of repelling precipitates during prolonged use. The higher eNSR activities of NCP/PC than those of CoP/PC, $Ni_2P/PC$, $MoS_2/PC$ (a higher $j$, a higher Tafel slope, a higher $C_{dl}$, and a lower $R_{ct}$, see Supplementary Fig. 46) may contribute to the optimal anti-precipitation performance. Indeed, NCP/PC demonstrates better gas release ability (Supplementary Fig. 32) than other PC-based cathodes (Supplementary Figs. 47–49). Note that single NCP, CoP, $Ni_2P$ and $MoS_2$ supported by CP also exhibit $H_2$ evolution activities in natural seawater (Supplementary Fig. 50), but all those electrodes present rather poor anti-precipitation abilities. In addition, enhancing electrode hydrophilicity is known to increase bubble-releasing speeds required to boost electrocatalytic activity. However, we found that the effect of hydrophilicity on the ability for an electrode to repel precipitates is limited. In other words, the hydrophilicity is not the decisive factor for NCP/PC to present the superior resistance to precipitation. For instance, massive insoluble precipitates formed and accumulated on the NCP/NF after the 10-h eNSR testing (Fig. 3f), whereas NCP/NF itself is superhydrophilic (Fig. 4j) featuring a water contact angle (WCA) of 0°. Moreover, both NCP/CC (WCA = 0°) and NCP/GF (WCA = 0°) were more hydrophilic than NCP/PC (WCA = 22.8°); however, the NCP/PC still performs far better in terms of the enhancement in $j$ (Fig. 3b), the eNSR stability (Fig. 3e), as well as the proficiency in repelling precipitates (Fig. 3f, g and Supplementary Figs. 26–28). Thus, hydrophilicity is not the main factor in the electrode resistance to precipitation. We recorded air contact angles (ACA) of all NCP-based samples (Fig. 4k) to understand why NCP/PC has the smallest average bubble diameters. The order of the samples with the ACA from large to small is: NCP/PC (151.12°) > NCP/NF (147.75°) > NCP/CP (147.17°) > NCP/CC (145.70°) > NCP/NF (147.75°) > NCP/TM (137.53°). With both WCA and ACA data, NCP/PC is thus the most hydrophilic and aerophobic electrode, which means that bubbles can easily detach during their formation processes. According to adhesive forces measurement data of gas bubbles on different NCP-based electrodes (Fig. 5a–f), the following trend indicates an increase in catalyst-bubble interfacial adhesion forces: NCP/PC (21.8 μN), NCP/NF (30.3 μN), NCP/CP (30.7 μN), NCP/CC (39.9 μN), NCP/GF (42.3 μN), NCP/TM (46.8 μN). The bubble-electrode adherence can lead to lower seawater reduction efficiency. The smallest adhesive force (21.8 μN) for NCP/PC again demonstrates that it should exhibit faster bubble

evolution kinetics. Based on the experimental observations, we propose a thorough summary of the likely causes for the best seawater reduction performance of NCP/PC in order to guide the future creation of more self-cleaning and robust eNSR cathodes, which is crucial for the maintenance of high electrode activity in low-grade and saline water with impurities (e.g., seawater) for prolonged tests.

As depicted in Fig. 5g–i, the primary superiority of NCP/PC is its unusual 3D $H_2$-evolving architecture, which brings vital benefits that boost electrocatalysis efficiency, facilitate $H_2$ gas release traffic and grant itself superb anti-precipitation ability. (1) The longitudinally arranged/elongated micro-channels with multiple side pores for the honeycomb-like PC framework (Fig. 1 and Supplementary Fig. 1) enable the favorable electrolyte ion permeation/accessibility and rich gas transport passages, which are quite unavailable in other NCP-based cathodes (Supplementary Fig. 51). We have performed a series of control experiments to explain the key roles of the pores of NCP/PC. As shown in Supplementary Figs. 52–54, the more damage is done to the pore structure of NCP/PC, the smaller ECSA and lower $j$ of the electrode (Supplementary Fig. 53), which indicates a decrease in the corresponding bubble release efficiency. We obtained the relationship between amounts of $Mg^{2+}/Ca^{2+}$ precipitates on the electrode and the related $j$ in Supplementary Figs. 54. The data do show a near linear relationship that the greater the $j$ (i.e., the less destroyed pore structure, as well as the more efficient bubble release), the less $Mg^{2+}/Ca^{2+}$ precipitates. (2) Only bubbles released by NCP/PC achieve the smallest average diameters and the most modest migration velocity, which may perfectly homogenize the force-repelling precipitates. (3) The reasonably thick PC with open and low-curved lumens also supports the growth of high-quality NCP. Note that PC is covered in dense and uniform nanostructured NCP, even in its aligned micro-channels, resulting in less dead materials and more readily accessible active sites. (4) The direct growth of NCP on the carbon as a binder-free cathode acquires strong carbon substrate-NCP adhesion and nice electron transfer. (5) Nanostructured NCP on PC, in contrast to slurry-coated planar electrodes, creates more contact sites with bubbles, which weakens adhesive forces of bubbles to the electrode and thus effectively encourages air bubble release. (6) The good hydrophilicity of NCP would further speeds up the detachment of smaller-size bubbles. Accordingly, small bubbles of similar sizes are efficiently released from every corner of the NCP/PC. (7) $H_2$ gas production under industrial-level $j$ within vessel channels may lead to localized pressures that force internal water/$H_2$ out via gas-accessible pores/holes at the bottom and sides. The surface of the NCP/PC will be further cleaned by such flows with high momentum, especially those ejected from the sides via extrusion. Except for a number of structural advantages, dissociation of water to $H_{ads}$ is thermodynamically preferred for the NCP/PC even in seawater (NCP as one of the most representative hydrogen evolution reaction (HER)-active materials covers the entire porous PC to offer densely/evenly distributed active sites for fast water dissociation), thus empowering our NCP/PC to sustain a high level of $\theta_H$ for massive $H_2$ gas release and subsequent repulsion of precipitates. Noticeably, the MBPTS with both 3D geometry features and superb intrinsic $H_2$-producing activity achieved by our NCP/PC can just be compared to advanced irrigation technologies, which evenly and precisely distributes the water needed by crops to the crop roots to maximum water use. Likewise, the NCP/PC uniformly and massively transports small-sized bubbles (like the water in irrigation) to every corner of the electrode (like the crop roots) to repel $Mg^{2+}/Ca^{2+}$ precipitates without a break, which fully improves the utilization of bubbles and boosts the efficacy of inhibiting cathode surface precipitation. In a nutshell, the unique architectures and fast eASR/eNSR kinetics of NCP/PC jointly achieve a micro-scale, consistent, and highly efficient mass transport system that methodically takes bubbles and precipitates away from the active sites. Reasons why the MBPTS works effectively are discussed more in Supplementary Note 3 and finite element calculation parts.

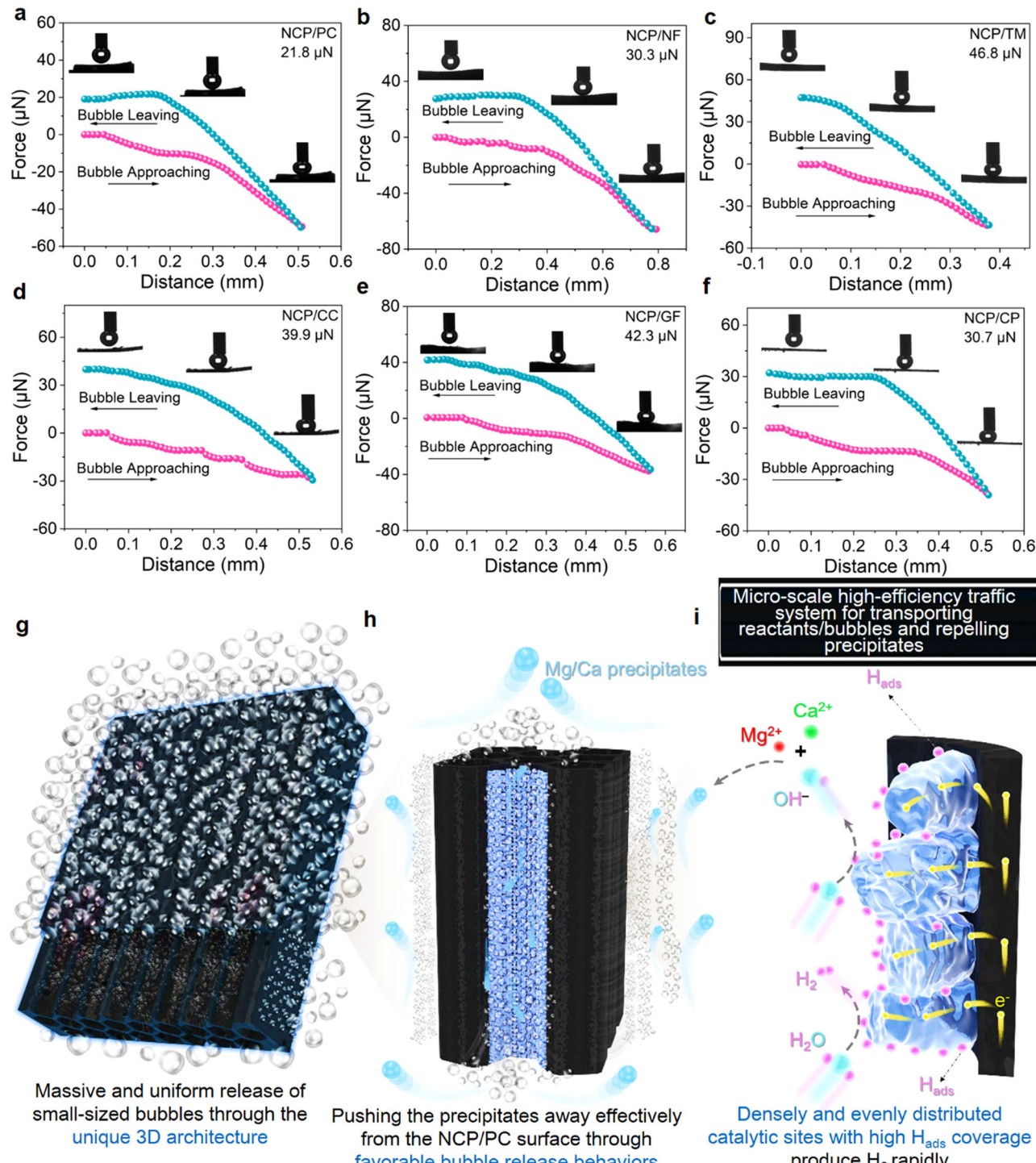

**Fig. 5 | Efficient transport/traffic micro-systems based on eNSR-active and anti-precipitation 3D architecture. a–f** Gas bubble adhesive force measurements of NCP/CP, PC, NCP/NF, NCP/TM, NCP/CC, NCP/GF, and NCP/CP. **g–i** Schematic diagrams of efficient traffic of bubbles and precipitates on/in the NCP/CP. The bubble acts as the cleaner to make NCP/CP free from precipitation by repelling precipitates without a break.

Major benefits of this work over previously published studies in terms of anti-precipitation seawater electrolysis are listed in Supplementary Note 4.

Finite element simulations involving two models of NCP/CP (i.e., the optimal sample) and the counterpart electrode provide more in-depth and comprehensive insights on the MBPTS. Methods of all simulating can be found in Supplementary Notes 5 and 6 and Supplementary Figs. 55 and 56. As the bubble diameter decreases, the related bubble slip velocity ($u_{gas}$-$u_{liquid}$, the velocity of the bubble

phase relative to liquid phase) decreases, but the pushing forces on the liquid phase are strengthened. The liquid velocity sharply increases when bubble diameter approaches a particular point (the tipping point), which causes the bubble velocity to increase rather than decrease (Supplementary Fig. 57a). Simulation results reflect the trends of such changes (Supplementary Fig. 57b). Since the bubble diameters of NCP/CP are much smaller than those of all other NCP-based samples, bubbles released by NCP/CP will be closer to the tipping point, resulting in the smallest bubbles with moderate moving

velocities. Bubbles are forced to attempt to cluster together by the gas-liquid interfacial tension to further reduce the overall surface area. This prevents bubbles from breaking free from the gas cluster. The detachment of individual bubbles (i.e., to break free from the gas cluster) thus requires a perturbation, which may originate from inhomogeneity in (1) pore geometry or (2) gas production density inside the electrode. The inhomogeneity in the liquid inlet velocity creates the perturbation (Supplementary Notes 6). The gas-liquid interface of the optimal sample is more sensitive to the perturbation, presenting the obvious convex (Supplementary Fig. 58a); however, the counterpart sample's pore connectivity is too strong (Supplementary Fig. 58b), making it hard for the perturbations to affect the gas-liquid interface and thus difficult to encourage detachment of bubbles. Moreover, the bubbles of the optimal sample coalesce only on the outer surface of the electrode, while the bubble of the counterpart sample has coalesced in it, which is another reason for the larger bubbles of the counterpart sample. We compared antiprecipitation abilities of different structures. Due to the directional movement of bubbles in the optimal sample, the precipitate can be discharged efficiently (Supplementary Fig. 59a); however, in the counterpart sample, due to the highly non-directional flow of the electrolyte solution (i.e., random directions), the precipitate discharge ability is not as good as that of the optimal sample (Supplementary Fig. 59b). Supplementary Fig. 60 further shows that vortices are easy to appear in the velocity field, this consists with the lower precipitate discharge ability stemmed from non-directional flow. Supplementary Fig. 61 compares the repelling ratio ($\eta$(t)) of the two electrodes, with lower $\eta$(t) for the counterpart sample. In addition, we tried to optimize the pores theoretically, and the corresponding results are given in Supplementary Figs. 62–65.

## Natural seawater-to-H₂ demonstration in a flow-type cell

We assembled a two-electrode electrolyzer with symmetric and flowing natural seawater feeding modes to further showcase practical uses of the 3D NCP/PC cathode (see Supplementary Note 7), which is schematically illustrated in Fig. 6a. Two chambers were separated by a proton exchange membrane (PEM), and the anode was a commercial dimensionally stable anode (DSA). Besides, pumps and seawater storage tanks were also employed in our flow-type electrolysis systems. Notably, Mg/Ca precipitates can hardly adhere to/cover the surface of our self-cleaning NCP/PC; instead, they would be transported out of the chamber by the flow as they sink. Since DSA directly splits natural seawater as well, active chlorine and $O_2$ were produced during the electrolysis, which can be discharged from the anodic chamber in time. Impressively, even with unprocessed natural seawater at 25 °C, robust electrocatalytic performance is achieved by NCP/PC||DSA (Fig. 6b), requiring the cell voltages of 2.65, 3.38, and 3.8 V at the $j$ of 200, 500, and 800 mA cm$^{-2}$. Moreover, the NCP/PC||DSA far outperforms the Pt/C/PC||DSA commercial counterpart, exhibiting higher $j$ in the identical voltage window. The actual $j$ distribution of a piece of NCP/PC (6 cm$^2$) during the eNSR catalysis was indirectly studied using *operando* infrared thermography (Supplementary Fig. 66). Although ample $Mg^{2+}$/$Ca^{2+}$ in natural seawater converts to precipitates near the NCP/PC, the strong anti-precipitation ability leaves the activity of the cathode completely undisturbed by insoluble precipitation. Our home-made flow-type electrolyzer catalyzed by the NCP/PC||DSA electrodes maintains a typical industrial operating $j$ of 500 mA cm$^{-2}$ for 150 h (Fig. 6c), during which there is a slight voltage loss (0.2893 mV h$^{-1}$). Such long-term seawater electrolysis durability is even better than the most advanced nature seawater electrolyzer (NSE) that operated at the $j$ of 500 mA cm$^{-2}$ for 100 h (notable voltage loss was observed, ~1 mV h$^{-1}$)[10]. The fact that the polarization curve virtually overlaps the curve recorded prior to the 150-h test (Fig. 6d), with no apparent activity decrease, further demonstrates the high stability of our NSE. As shown in the inset table of Fig. 6c, the energy consumption for the

NCP/PC||DSA-based NSE is 90.6 kWh kg$^{-1}$ H₂ at −0.5 A cm$^{-2}$. The US Department of Energy objective of 2.0 US\$ kg$_{H2}^{-1}$ is higher than the anticipated electricity cost produced by our NSE[66], which is roughly 1.8 US\$ kg$_{H2}^{-1}$ (see details in Supplementary Note 8). Please take notice that despite the fact that we admit the limitations of our electrolyzer assembly/construction techniques, the electrochemical stability of our NCP/PC||DSA electrodes still stands out from the reported two-electrode natural seawater electrolyzers (Supplementary Table 7). Additionally, for mild seawater-to-H₂ electro-conversion systems to be implemented in practice, large $j$, reliable electrolysis operation, and high product selectivity are all essential. Importantly, the H₂ FE values are consistently maintaining at close to 100% over the course of the prolonged operation when the $j$ is fixed industrial-level 500 mA cm$^{-2}$ (Fig. 6e and Supplementary Fig. 67), demonstrating the success of our 3D NCP/PC cathodes in retaining both high eNSR activity and selectivity during extended periods of electrolysis time. We used ICP-AES to estimate the elemental loss of NCP/PC. As shown in Fig. 6f, P loss is far greater than Ni loss and Co loss, and none of the three fluctuate much after ~15 h, implying that NCP on PC should transition to a new active Ni- and Co-based metastable state. XPS and infrared spectroscopy data of the NCP/PC after the electrolysis (Supplementary Figs. 68–70) confirm the loss of surface P and the chemical state changes in Co and Ni species. Moreover, clear characteristic peaks that should belong to M−O bonds (M = Ni or Co) in (oxy)hydroxides appear at potentials more negative than −1.5 V, as proven by in situ spectroscopic data for NCP/PC (see the details in Supplementary Figs. 28), thus suggesting the surface phosphide-to-(oxy)hydroxides conversion. As a matter of fact, during HER electrolysis, it is fairly normal for the initial states of metal phosphides to experience reconstruction (Supplementary Note 9). Most importantly, metal phosphide transformations are not the emphasis of this work, and NCP is only one example for illustrating the PC-based MBPTS design. We reason that P losses affect little the eASR/eNSR performance, given previous findings (see Supplementary Note 9) as well as the stable potential-current responses in prolonged tests in our work (Figs. 2d, 6c and d). The potential of NCP/PC for industrially relevant eNSR applications are further explored in a scaled-up seawater electrolyzer consisting of multiple single stacks, which demonstrates 10-h durability (Supplementary Fig. 71).

In summary, we discover/investigate electrodes that combine merits of honeycomb-like carbon skeleton and H₂-evlution nanocatalyst for high-efficient, robust and $Mg^{2+}$/$Ca^{2+}$ precipitation-less eASR/eNSR. Our electrodes have an architecture rich in transmission highways that transfer electrolytes/bubbles methodically and simultaneously to repels precipitates. Bubbles released by NCP/PC achieve the most modest migration velocity and the smallest average diameter, which homogenize the forces repelling precipitates. Importantly, different catalytic materials also obtain generally improved precipitation resistance after growing onto the PC (see Supplementary Fig. 45), indicating that our PC-based MBPTS design is of universal significance. With a small Tafel slope and noticeably low overpotentials, NCP/PC represents state-of-the-art developments in eASR cathodes. It also operates at the ampere-level current density of −1 A cm$^{-2}$ for at least 1000 h without failure, far more superb than any previous cathodes. Moreover, NCP/PC enables the greatest anti-precipitation ability among all the cathodes, with no visible surface white precipitates and a low $A_{Ca\&Mg}$ value after the 10-h eNSR operation under −0.5 A cm$^{-2}$. Additionally, the flow-type electrolyzer with NCP/PC||DSA stably functions at 500 mA cm$^{-2}$ for 150 h without any discernible attenuation, while sustaining high H₂ FE values (above 96.5%) throughout the 150 h. To our best knowledge, this represents one of the most robust overall natural seawater electrolysis performance. This work not only demonstrates uniquely constructed 3D cathodes with record-high eASR and eNSR performances but also reveals universal principles underlying superb anti-precipitation performances (i.e., the

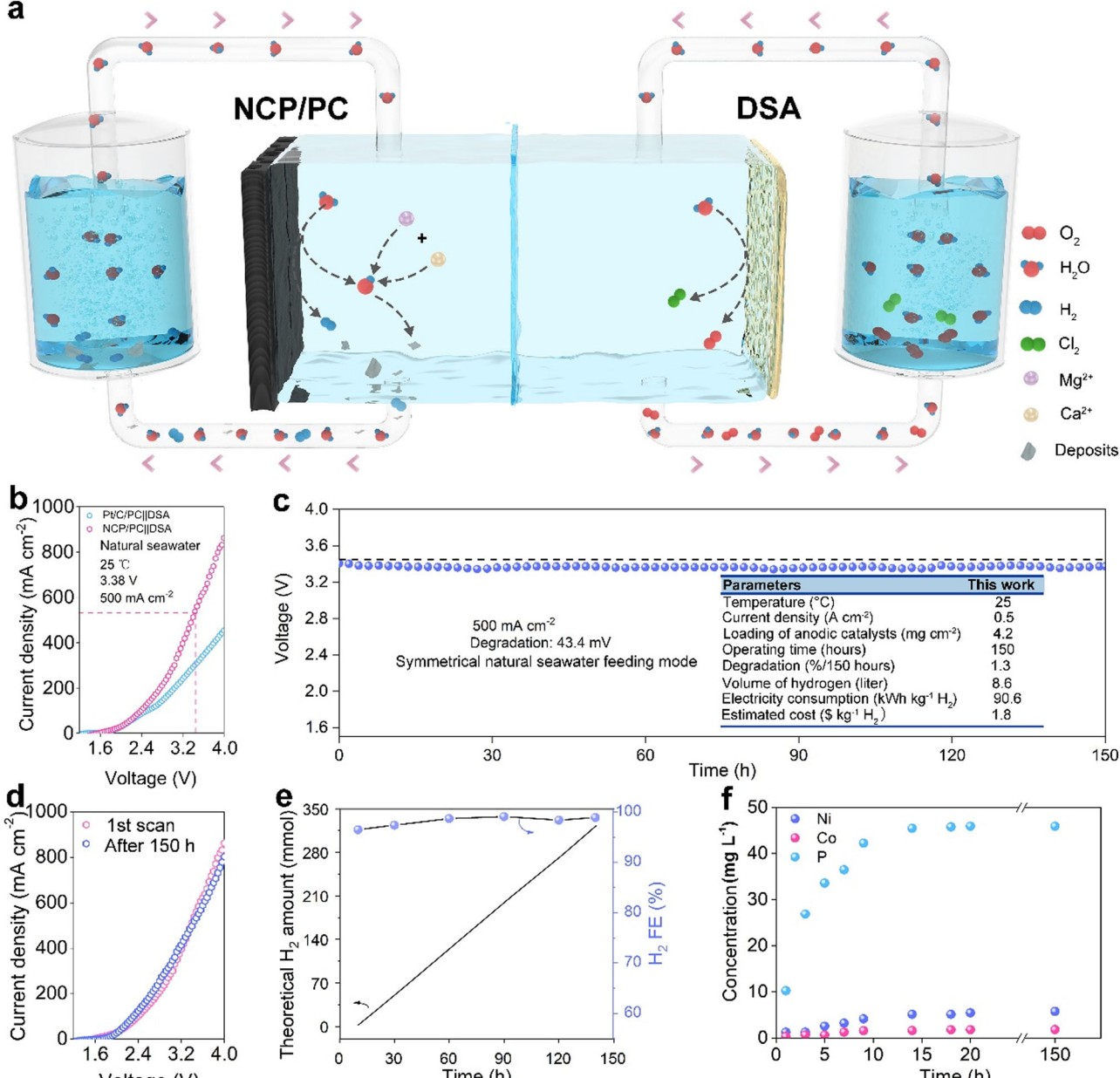

**Fig. 6 | Overall natural seawater electrolysis performance of two-electrode system in a flow-type electrolyzer. a** Schematic diagram of the overall structure of flow cell. The purple and light brown spheres in the cathode chamber represent $Mg^{2+}$ and $Ca^{2+}$, respectively, which capture $OH^-$ to form the precipitates (represent by a gray fragment). In the anode chamber, $O_2$ gas (the red molecules) and some active chlorine (e.g., $Cl_2$, the green molecules) are formed in situ on the commercial dimensionally stable anode (DSA). **b** Polarization curves for NCP/PC||DSA and Pt/ PC||DSA at 25 °C (without *iR* correction). **c** Long-term stability of NCP/PC||DSA under a fixed *j* of 500 mA cm$^{-2}$. The overall degradation rate: 0.2893 mV S$^{-1}$ (without *iR* correction). **d** Almost unchanged polarization curves before and after the durability testing at 25 °C. **e** $H_2$ FE throughout the test and the theoretical amount of $H_2$ produced. **f** The leaching amounts of the phosphides during the durability test. The catalyst mass loading for cathode is ~0.0042 g cm$^{-2}$. The pH of the electrolyte is 7.89. The charge transfer resistance of NCP/PC is 9.0 Ω.

PC-based MBPTS design) to solve the long-standing issues in seawater electroreduction, which provides vital insights for electrode designs toward industry electrolysis of seawater.

## Methods

### Fabrication of the PC
Peeled natural pine wood was first cut into the required sizes (~2 cm × ~3 cm × ~0.3 cm), which were then submerged for 1 h in a solution of 1 M hydrochloric acid. Next, the wood blocks were repeatedly rinsed with ethanol and ultrapure water until the pH of the water was about 7.0. The wood blocks were dried in a vacuum oven (DZF-6050, Shanghai Qixin Scientific Instrument Co., Ltd) under 60 °C for 12 h.

After drying, the wood samples were heated in a tube furnace (GSL-1700X, Hefei Kejing Material Technology Co., Ltd.) at 1000 °C (the heating rate: 3 °C per minute) for 6 h under Ar atmosphere (purity: over or equal to 99.999%) to obtain the PC. The carbonized pine wood was further cleaned and dried for electrochemical testing or for further catalyst loading.

### Fabrication of the NCP/PC and five other NCP-based cathodes
In 38 mL of ultrapure water, 0.4754 g of NiCl$_2$•6H$_2$O, 0.9517 g of CoCl$_2$•6H$_2$O, and 0.4504 g of urea were dissolved and thoroughly mixed before being transferred to a 50 mL Teflon-lined stainless steel autoclave. Then place a piece of PC in the autoclave for further

hydrothermal reaction (120 °C for 6 h) in an electric thermostatic air-drying oven (DHG-9075A, Shanghai Qixin Scientific Instrument Co., Ltd). After cooling to ~30 °C, the hydrothermally prepared NiCo precursor/PC was taken out from the autoclave, cleaned with ultrapure water and ethanol, and dried in a vacuum oven. Nanostructured NCP on the honeycomb-like PC was obtained via the facile topotactic phosphidation of the NiCo precursor/PC at 300 °C in the Ar for 2 h, using $NaH_2PO_2$ as the $PH_3$ source. The other NCP-based samples were synthesized in the same way as NCP/PC except for the substrate used. NF was soaked in 1 M hydrochloric acid for 20 min, then washed repeatedly with ethanol and ultrapure water, and finally dried for use. Both GF and CC were soaked in heated concentrated nitric acid for 2 h, then cleaned repeatedly with ethanol and ultrapure water and finally dried for later use. TM was soaked in 1 M hydrochloric acid for 1 h, then washed repeatedly with ethanol and ultrapure water, and finally dried for use. CP was used after cleaning with acetone, ethanol and water and then dried.

### Fabrication of the $Ni_2P/PC$ and CoP/PC
The synthesis of monometallic metal phosphides was similar to that of NCP/PC, but only the corresponding metal salt was added.

### Characterization
The crystal phase structures for all samples were examined by an X-ray diffractometer (DANDONG HAOYUAN INSTRUMENT Co., Ltd., Cu Kα radiation) with a Dwlc-3 Water circulating system. The catalyst surface chemistry was investigated using XPS (Thermo Scientific TM Nexsa TM, USA) with a monochromatic X-ray source (Al Kαhυ = 1486.6 eV). Gemini SEM 300 electron microscope (ZEISS, Germany) combined with EDX was employed to record the morphologies and element composition/distribution of the samples. TEM and HRTEM images were obtained on a JEM-F200 electron microscope (JEOL Ltd.). The mass loadings of NCP on various substrates (e.g., CC) were calculated based on the results obtained on an electronic analytical balance (SHIMADZU, AUW220D). The contents of Ni and Co from the leaching/dissolution of NCP/PC in the seawater overtime during the reaction were determined by ICP-AES (iCAD7400). The behaviors of gaseous $H_2$ and precipitates on the cathodes were recorded by a high-speed camera (revealer, m230/mm220, HF Agile Device Co., Ltd.). All WCA data were measured by using a video-based contact angle measuring device (KRUSS DSA100).

### Electrochemical measurements
Natural seawater was taken from Huangdao District, Qingdao City, Shandong Province, China, and ion compositions of the natural seawater (before the seawater alkalization) and alkaline seawater (after the seawater alkalization) are listed in Supplementary Table 8. No filtration treatment of seawater was performed prior to overall natural seawater electrolysis in Fig. 6, as the collected seawater itself was clear (Supplementary Fig. 72). The process of preparing alkaline seawater involved the use of a microporous filtration membrane (pore size: 0.45 μm) and a water-circulation multifunctional vacuum pump for filtration. All eASR and eNSR tests were carried out on three electrochemical workstations (CHI1140C, CHI 760E C20566, and CHI 600E A15663b). The eASR performance was recorded with a three-electrode system made up of the working electrode (i.e., PC, CoP/PC, $Ni_2P/PC$, NCP/PC, benchmark commercial Pt/C/PC, etc.), Hg/HgO (1 M KOH) as the reference electrode, and a commercial graphite rod (the diameter: 6 mm, the length: 150 mm) as the counter electrode. The mass loading for the Pt/C/PC was ~0.0042 g cm$^{-2}$. For eNSR experiments, the reference electrode was changed to a saturated Ag/AgCl electrode. A customized cell (Gaoss Union Optoelectronic Technology Co., Ltd) was used for all three-electrode system measurements. Linear scan voltammetry (LSV) was performed at a scan rate of 10 mV s$^{-1}$.

ECSA and ECSA-normalized $j$. A greater value of $C_{dl}$ generally indicates a higher ECSA for the electrochemistry reaction on the electrode. All ECSA values of our cathodes were obtained by dividing the $C_{dl}$ (mF cm$^{-2}$) by a fixed specific capacitance ($C_s$, μF cm$^{-2}$). Since the $C_s$ values of metal phosphide-containing materials were typically 40 μF cm$^{-2}$, we chose 40 μF cm$^{-2}$ to calculate the ECSA[67–69].

Definition of accumulation of Ca&Mg (mg cm$^{-2}$). Since the Mg/Ca-based precipitates build up on the cathodes to affect the efficiency of $H_2$ production in seawater reduction seriously, we quantified the accumulation of in situ formation precipitates by detecting elemental Ca&Mg on the post-eNSR cathodes. The post-eNSR cathodes were soaked in dilute hydrochloric acid solution for up to 12 h to obtain solution with $Mg^{2+}$ and $Co^{2+}$ for the accurate determination. We quantified the $A_{Ca\&Mg}$ values using the following Eq. (1):

$$A_{Ca\&Mg} = m_{Ca\&Mg}/GA \tag{1}$$

where $m_{Ca\&Mg}$ is the mass of Ca and Mg on the cathode obtained from ICP-AES data, GA is the geometric area of the electrode (cm$^2$).

Flow-type cell assembly and testing. Since the cathode can expel the precipitates, we used flowing seawater to avoid the accumulation and blockage of the precipitation in the electrolytic cell. We used the commercial anode for flow-type cell tests to eliminate any possible attenuation of anode activity that might affect the overall stability. Additionally, we employed digital peristaltic pumps (CHONRY) to reliably drive the seawater on both sides of the electrolyzer to form the loop in the electrolytic chamber, pipeline and seawater storage tank. As the electrolysis proceeded, the precipitates in the seawater storage tanks progressively accumulated, but these precipitates will gently sunk to the bottom of the tanks and remained there.

$H_2$ FE measurements. The typical drainage method is used to test determine the FE.

## Data availability
The data supporting the findings in this study are available within the paper and its Supplementary Information. Source data are provided with this paper.

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

## Acknowledgements

B.T. acknowledges the funding support from the Natural Science Foundation of China (No. 21927811). X.S. acknowledges the funding support from the Natural Science Foundation of China (No. 22072015) and the Free Exploration Project of Frontier Technology for Laoshan Laboratory (No. 16-02). Nanqiao Liu from SHENZHEN ZHONOYI TESTINO TECHNOLOGY SEAVICE CO., LTO is acknowledged for his support in discussions and theoretical simulations.

## Author contributions

X.S., J.L. and Z.C. designed this research. J.L. and Z.C. performed material synthesis, characterizations, and electrocatalysis tests. J.L. and Z.C. analyzed the experimental and theoretical data. J.L. and Z.L. conceived and completed all the schematic drawings. Y.L. and D.Z. contributed to the SEM measurements. Y.Y., S.S., and Q.L. participated in discussions. J.L. wrote the paper. X. S. and B. T. supervised the research. All authors contributed and reviewed the manuscript.

## Competing interests

The authors declare no competing interests.
