## [Peer Review File · Nature Communications]

REVIEWER COMMENTS

Reviewer #1 (Remarks to the Author):

Sun and co-workers report on a well-designed electrode architecture with a microscopic bubble/precipitate traffic system which exhibits genuine anti-precipitation properties during seawater reduction electrolysis. First and foremost, the formation of Ca^{2+} and Mg^{2+} precipitates on the surface of the electrode is a long-standing and unresolved issue in seawater electrolysis (Energy Environ. Sci., 11, 1898–1910 (2018), Nat. Energy 5, 367–377 (2020). Nat. Commun. 14, 3934 (2023)), well known to the community and is certainly challenging to deal with. Surprisingly, the NCP/PC electrode reported by the authors were not covered by $\text{Ca}^{2+}/\text{Mg}^{2+}$ precipitates at all after the long-term operation under industrial-level current densities in natural seawater, but were as clean as they were before electrolysis (Fig. 3f). Necessarily, the work under consideration should be regarded of high significance to the field of catalytic electrolysis of low-grade water like seawater. Furthermore, the results of high-speed camera observations are quite convincing in demonstrating the importance of the microscopic bubble/precipitate traffic system as a crucial design in the future investigation of precipitation-less seawater reduction.

Additionally, the record-high electrolysis performances, such as the stable operations lasting 1000 h in alkaline seawater under 1 A/cm² and 150 h in natural seawater under 0.5 A/cm², as well as the low estimated price of 1.8 US\$/kg for H₂ generation, are all impressive. Overall, this is a nice and highly informative work supported by solid data, which is both fundamentally important and practically useful towards the seawater electrolysis applications. Therefore, I do recommended this work to be accepted after a few minor revisions. Detailed points are outlined below.

- 1) In Fig. 3d, the corresponding specific values are suggested to be provided in the Supplementary Information.
- 2) The precipitation amount on the surface of NCP/CP is the largest (Fig. 3d), and whether the thickness of the electrode can significantly affect the formation of surface precipitation.
- 3) How the FE values for the flow-type electrolytic cell were determined throughout the long-term tests?
- 4) A few typos in the text should be corrected.
- 5) The flow-type electrolyzer with the anti-precipitation cathode revealed exceptional stability up to 150 h at 500 mA cm⁻², using symmetric natural seawater feeds. More discussions on the symmetric natural seawater electrolysis in this work would be beneficial to guide future work since there are fewer studies that can achieve symmetrical overall natural seawater splitting under the industrially-relevant current densities like 500 mA cm⁻² (Nat Energy 8, 264–272 (2023)).
- 6) In addition, the authors should elaborate more on the advantages of this work over previously reported work in terms of anti-precipitation seawater electrolysis.

Reviewer #2 (Remarks to the Author):

In this manuscript, the authors proposed a vital microscopic bubble/precipitate traffic system (MBPTS) by constructing honeycomb-type 3D cathodes for robust anti-precipitation seawater reduction. The constructed NiCoP/PC (NCP/PC) cathode with a honeycomb-like porous carbon framework rich in non-random open micro-channels shows excellent hydrogen evolution activity and stability in both alkaline seawater and natural seawater. In general, this manuscript highlights the structural advantages of pinewood-derived carbon (PC) framework for anti-precipitation performance during HER in natural seawater. I think this manuscript is suitable for publication in Nat. Commun after major revision by addressing the following concerns:

1. More details about the seawater used in the experiment should be provided, including the source, and the $\text{Ca}^{2+}/\text{Mg}^{2+}$ ion concentration before and after seawater alkalization, etc.
2. The authors emphasized the important role of PC substrate in the anti-precipitation performance of electrode materials, and more structural analysis of PC materials should be provided.
3. The authors demonstrated that the NCP/PC achieve the smallest average diameter and the most modest migration velocity. The authors should clarify this issue carefully by experiments and theoretical evidences.
4. As shown in Figure 3f, the authors emphasize that the massive insoluble precipitates formed and accumulated on the NCP/NF after the 10-h eNSR testing, but these pictures provided in the manuscript are difficult to support each component and structure. The author needs to clarify this issue carefully by in-situ evidences.
5. In order to better clarify the anti-precipitation advantage of the PC structure, a comparison of the ECSA changes of NCP with different substrates (CP, CC, GF, TM, and NF) after 10 h electrolysis should be provided.
6. Electronic metal–support interactions (EMSI) modulation mechanism of NCP on various substrates should be given. The authors may need to describe the reaction mechanism in more details. Discuss clearly importance of NCP and correlation to the substrates.
7. A series of in situ and ex situ characterizations to investigate the effect of transport/traffic micro-systems on the structural stability of NCP on various substrates for HER in seawater systems. Meanwhile, the author may consider establish the approximate linear relationship of $\text{Mg}^{2+}/\text{Ca}^{2+}$ precipitates and efficient traffic of bubbles.
8. The authors claim that NCP/PC holds better HER performance compared to other PC electrodes (CoP/PC, Ni₂P/PC, MoS₂/PC), which may contribute to the optimized anti-precipitation ability, while a comparative analysis of their performance as well as single NCP, CoP, Ni₂P and MoS₂ should be provided.
9. The real-time high-definition images provided seem to be difficult to obtain direct evidence about the microscopic bubble/precipitate traffic system (MBPTS), which is also the main highlight of the manuscript, so clearer hints and key evidence need to be given.

10. Optical photographs of the Faradaic efficiency test also need to be provided.

Reviewer #3 (Remarks to the Author):

Sun and co-workers described an emerging 3D micro/nano-structured electrodes that combine merits of honeycomb-like carbon skeleton and H₂-evolution nanocatalyst for eASR/eNSR. Bubbles released by NCP/PC achieve the most modest migration velocity and the smallest average diameter. It also operates at the ampere-level current density of -1 A cm^{-2} for at least 1000 h without failure. The flow-type electrolyzer with NCP/PC || DSA stably functions at 500 mA cm^{-2} for 150 h without any discernible attenuation, while sustaining high H₂ FE values (above 96.5%) throughout the 150 h. This is a routine addition to a well-known field. This manuscript suffers from following lacks:

1. The areal current density is divided by the geometric area of the electrode. I noticed the size of electrode is quite small, and the thickness of the electrode is very large. Therefore, it is very easy to obtain very large current density (e.g. 500 mA cm^{-2}) in lab. In an industrial electrolyzer, the electrode is very large and the distance between cathode/anode is very small. The foam is applied in most cases, but its thickness is very small. Therefore, the validation of very large current density at an industrial electrolyzer is missing now.
2. The actual seawater should be applied and demonstrated if this work is claimed as ultrastable seawater reduction electrocatalysis.
3. How to regulate the bubble size herein? The match between the pore size and operating current density is quite important.
4. The evolution of catalyst surface functional groups should be demonstrated. It is widely accepted that the surface of many electrocatalysts has been oxidized/reduced into active states.
5. The diffusion of feedstocks/products from micro/mesopores to the bulk electrolyte should be quantitatively analyzed.
6. The mechanical and electrical conductivity of 3D micro/nano-structured electrodes should be further checked. The optimization of pore size, pore thickness, and areal loading should be explained.
7. How to anchor taps on the structured electrocatalysts? What is the actual current density distribution and related gas evolution?
8. What is the maximum current density on the electrocatalysts in an industrial device?

Based on these considerations, the current manuscript is less strong for a high impact journal like Nature Commun. It can be reconsidered by Commun Mater or Commun Chem.

Point-by-Point Responses to Reviewers' Comments

We sincerely thank the editor and all reviewers for their valuable feedback that we have used to improve the quality of our manuscript (NCOMMS-23-44028). The reviewer comments are shown in *italic* and **bold**, and our responses, including the added figures, tables, descriptions, and other items, are highlighted in the **blue** text.

Point-by-point response to the reviewers #1

Reviewer #1 (Remarks to the Author): Sun and co-workers report on a well-designed electrode architecture with a microscopic bubble/precipitate traffic system which exhibits genuine anti-precipitation properties during seawater reduction electrolysis. First and foremost, the formation of Ca^{2+} and Mg^{2+} precipitates on the surface of the electrode is a long-standing and unresolved issue in seawater electrolysis (Energy Environ. Sci., 11, 1898–1910 (2018), Nat. Energy 5, 367–377 (2020). Nat. Commun. 14, 3934 (2023)), well known to the community and is certainly challenging to deal with. Surprisingly, the NCP/PC electrode reported by the authors were not covered by $\text{Ca}^{2+}/\text{Mg}^{2+}$ precipitates at all after the long-term operation under industrial-level current densities in natural seawater, but were as clean as they were before electrolysis (Fig. 3f). Necessarily, the work under consideration should be regarded of high significance to the field of catalytic electrolysis of low-grade water like seawater. Furthermore, the results of high-speed camera observations are quite convincing in demonstrating the importance of the microscopic bubble/precipitate traffic system as a crucial design in the future investigation of precipitation-less seawater reduction.

Additionally, the record-high electrolysis performances, such as the stable operations lasting 1000 h in alkaline seawater under 1 A/cm^2 and 150 h in natural seawater under 0.5 A/cm^2 , as well as the low estimated price of 1.8 US\$/kg for H_2 generation, are all impressive. Overall, this is a nice and highly informative work supported by solid data, which is both fundamentally important and practically useful

towards the seawater electrolysis applications. Therefore, I do recommended this work to be accepted after a few minor revisions. Detailed points are outlined below.

General Response: The reviewer's recommendation for acceptance of our work is sincerely appreciated, and we thank him/her for finding our paper to be of significance and high quality. According to suggestions, we have made corrections to our previous draft, and the specifics are provided below.

Comment 1: In Fig. 3d, the corresponding specific values are suggested to be provided in the Supplementary Information.

Response 1: As suggested, we have provided the corresponding specific values in the revised Supplementary Information (please see **Table R1**).

Table R1. C_{dl} , mass activity, and η for various cathodes in natural seawater.

Electrode	C_{dl}	mass activity	η
NCP/PC	86.1 mF cm ⁻²	26.733 A mg ⁻¹ @-0.5 V _{RHE}	553.8 mV at 500 mA cm ⁻²
NCP/NF	72.1 mF cm ⁻²	2.689 A mg ⁻¹ @-0.5 V _{RHE}	874 mV at 500 mA cm ⁻²
NCP/TM	68.6 mF cm ⁻²	3.459 A mg ⁻¹ @-0.5 V _{RHE}	961.4 mV at 500 mA cm ⁻²
NCP/CC	35.5 mF cm ⁻²	4.855 A mg ⁻¹ @-0.5 V _{RHE}	1116.4 mV at 500 mA cm ⁻²
NCP/GF	55.2 mF cm ⁻²	2.348 A mg ⁻¹ @-0.5 V _{RHE}	992.6 mV at 500 mA cm ⁻²
NCP/CP	30.4 mF cm ⁻²	4.322 A mg ⁻¹ @-0.5 V _{RHE}	1213.3 mV at 500 mA cm ⁻²

Comment 2: The precipitation amount on the surface of NCP/CP is the largest (Fig. 3d), and whether the thickness of the electrode can significantly affect the formation of surface precipitation.

Response 2: We appreciate this comment. In general, the quantity of surface precipitate is not greatly impacted by electrode thickness. In comparison to NCP/TM or NCP/CC, for example, NCP/NF is substantially thicker, but it has more precipitates accumulated on it than those on the NCP/TM or NCP/CC. The amount of Mg²⁺/Ca²⁺ precipitation is actually determined by the electrode's anti-precipitation capabilities. In our revised

Manuscript and Supporting Information, the anti-precipitation ability of NCP/PC is further explained with more experimental and theoretical data.

Comment 3: How the FE values for the flow-type electrolytic cell were determined throughout the long-term tests?

Response 3: We appreciate this comment. We conducted the gas quantification tests at regular intervals and determined the each FE value in the selected range (25-min testing for each measurement). The H₂ gas produced from the cathodic chamber will be fed into the drainage tube. The optical photographs of the FE tests can be found in the revised **Supplementary Fig. 67**.

Comment 4: A few typos in the text should be corrected.

Response 4: As suggested, we have corrected the typos in our revised manuscript and Supporting Information.

Comment 5: The flow-type electrolyzer with the anti-precipitation cathode revealed exceptional stability up to 150 h at 500 mA cm⁻², using symmetric natural seawater feeds. More discussions on the symmetric natural seawater electrolysis in this work would be beneficial to guide future work since there are fewer studies that can achieve symmetrical overall natural seawater splitting under the industrially-relevant current densities like 500 mA cm⁻² (Nat Energy 8, 264–272 (2023)).

Response 5: Thanks for raising this. In fact, any feeding mode (either asymmetric one-side feed or symmetric two-side feeds) can be adopted for seawater electrolysis (please refer to: *Mater. Today* **69**, 193–235 (2023), *Nat. Energy* **5**, 367–377 (2020)). In the field of seawater electrolysis, there are also notable advances in symmetric electrolysis (please refer to: *Nat. Commun.* **14**, 3607 (2023), *Nat. Energy* **8**, 264–272 (2023)), of which the use of natural seawater on both sides are less because when untreated natural seawater is supplied as the electrolyte to both sides, the anode would suffer from severe chlorine-based corrosion, while the cathode surface would be physically covered by severe Mg²⁺/Ca²⁺-based precipitation (i.e., more dead/blocked reaction sites). Moreover,

asymmetric electrolysis can avoid one of these situations directly to a certain extent (please refer to: *Energy Environ. Sci.* **13**, 1725-1729 (2020), *Adv. Mater.* **33**, 2101425 (2021), *Small* **19**, 2208076 (2023)). Different feeding modes have the own advantages, and it is necessary to adopt a suitable feeding method according to a specific testing situation, performance/cost requirements, or the electrolyzer components, *etc.* The above discussions can be found in **Supplementary Note 7**.

Comment 6: *In addition, the authors should elaborate more on the advantages of this work over previously reported work in terms of anti-precipitation seawater electrolysis.*

Response 6: As suggested, we have elaborated more on the advantages of our work over previously reported work in terms of anti-precipitation seawater electrolysis. The related discussions can be found in **Supplementary Note 4** (can also be found below).

The primary obstacle to direct electroreduction of natural seawater is the continuous formation of insoluble $\text{Mg}^{2+}/\text{Ca}^{2+}$ precipitates on the cathodic electrode, particularly when the operating current densities are high (please refer to: *Nat. Energy* **5**, 367–377 (2020)). Moreover, even when using alkaline seawater, the residual Mg^{2+} and Ca^{2+} will still continue to accumulate and form tiny precipitates on the surface of the catalyst over a long period of time, ultimately leading to degradation of performance. Therefore, if seawater needs to be used as electrolytes for generation of H_2 , the ways to repel $\text{Mg}^{2+}/\text{Ca}^{2+}$ precipitates should be discovered. A study by Zeng et al. focused on improving the catalyst-substrate interaction for longer-term stability in eNSR (please refer to: *J. Mater. Chem. A* **7**, 25628–25640(2019)). Anchoring single-atom Pt on CoP also outperformed the Pt/C counterpart for long-term eNSR (please refer to: *J. Mater. Chem. A*, **8**, 11246–11254 (2020)). A core-shell NiMo@C₃N₅/glassy carbon electrode also showed better eNSR stability than Pt/C (please refer to: *Chem. Eng. J.* **438**, 135379 (2022)). Although the catalysts showed HER activities in natural seawater, the stability is not ideal due to the problem of precipitation of $\text{Mg}^{2+}/\text{Ca}^{2+}$. For example, while the NiMo wrapped by C₃N₅ shells can be protected from the poison of impurities, white insoluble salts were constantly deposited on the NiMo@C₃N₅/glassy carbon electrode

(please refer to: *Chem. Eng. J.* **438**, 135379 (2022)).

To the best of our knowledge, no HER electrode shows particularly high stability in natural seawater. In stability measurements, the previously reported HER electrodes generally displayed current densities below 100 mA cm^{-2} and electrolysis periods under 100 h (please refer to: *J. Mater. Chem. A* **7**, 25628–25640 (2019), *J. Mater. Chem. A* **8**, 11246–11254 (2020), *Chem. Eng. J.* **438**, 135379 (2022), *Adv. Energy Mater.* **9**, 1901333 (2019), *Appl. Catal. B* **304**, 120993 (2022), *J. Electroanal. Chem.* **916**, 116379 (2022), *Green Chem.* **23**, 4551–4559 (2021), *J. Mater. Chem. A* **8**, 25768–25779 (2020), *ACS Energy Lett.* **5**, 2681–2689 (2020), *Energy Environ. Sci.* **10** 788–798(2017), *J. Energy Chem.* **55** 92–101 (2021), *ACS Nano*, **12**, 12761–12769 (2018), *Nat. Commun.* **13**, 5785 (2022)). While these studies proposed various catalysts capable of electrolysis in natural seawater, few of them targeted the design of precipitation-repelling catalysts. Recently, a novel electrode design for precipitation-free natural seawater reduction that relies on a special local surface micro-environment was recently reported by Guo et al. (please refer to: *Nat. Energy* **8**, 264–272 (2023)). They proposed that the strong binding of OH^- on Cr_2O_3 would restrict the generated OH^- within the electrical double layer. Although this work captured scientific novelty, the electrode's capacity for preventing precipitate formation was limited, with a precipitation-free electrolysis demonstration at the current density of 100 mA cm^{-2} for 2 h. Our work not only demonstrates uniquely constructed 3D cathodes with record-high natural seawater reduction performances but also reveals universal principles underlying superb anti-precipitation performances (i.e., the PC-based MBPTS design) to solve the long-standing issues in seawater electroreduction, which provides vital insights for electrode designs toward industry electrolysis of seawater.

Point-by-point response to the reviewers #2

Reviewer #2 (Remarks to the Author): In this manuscript, the authors proposed a vital microscopic bubble/precipitate traffic system (MBPTS) by constructing honeycomb-type 3D cathodes for robust anti-precipitation seawater reduction. The constructed NiCoP/PC (NCP/PC) cathode with a honeycomb-like porous carbon framework rich in non-random open micro-channels shows excellent hydrogen evolution activity and stability in both alkaline seawater and natural seawater. In general, this manuscript highlights the structural advantages of pinewood-derived carbon (PC) framework for anti-precipitation performance during HER in natural seawater. I think this manuscript is suitable for publication in Nat. Commun after major revision by addressing the following concerns:

General Response: Great thanks for your supportive feedback on our manuscript. We appreciate your time and effort in reviewing our work and providing professional and invaluable suggestions. As you are concerned, there are several problems that need to be addressed. We have followed your suggestions to further improve the quality of our manuscript. The detailed corrections are listed below.

Comment 1: More details about the seawater used in the experiment should be provided, including the source, and the $\text{Ca}^{2+}/\text{Mg}^{2+}$ ion concentration before and after seawater alkalization, etc.

Response 1: Thanks for raising this. As suggested, we have provided more details of the seawater used in the experiment (please see **Table R2**), including the source, the pH values, and the $\text{Ca}^{2+}/\text{Mg}^{2+}$ ion concentrations before and after seawater alkalization, *etc.* Moreover, we have provided the photos of seawater collected (please see the seawater in **Fig. R1**). Since the seawater itself is clean enough (no notable suspended/floating particles), we used it directly for electrolysis demonstration without filtering it. Note that previous natural seawater electrolysis work (please refer to: *Nat. Energy* **8**, 264–272 (2023)) used filtered natural seawater, while this work did not filter natural seawater, allowing for genuine direct natural seawater reduction.

Table R2. Ion compositions and pH values of the natural seawater (before the seawater alkalization) and alkaline seawater (after the seawater alkalization) used in this work.

Species	Conc. [mg L^{-1}] for natural seawater	Conc. [mg L^{-1}] for alkaline seawater
Mg^{2+}	1016.751	0.16
Ca^{2+}	358.04	8.371
K^{+}	358.04	35123.26
SO_4^{2-}	658.86	664.689
Na^{+}	9877.836	11492.33

Notes: The seawater was collected from Huangdao district, Qingdao city, China (in summer). The pH value for natural seawater was around 7.89, and pH for alkaline seawater was around 13.88.

Fig. R1 Natural seawater directly collected from the Yellow Sea, China. Note that the seawater can be used directly as the electrolyte for eNSR measurements in this study without disinfection or filtration.

Comment 2: The authors emphasized the important role of PC substrate in the anti-precipitation performance of electrode materials, and more structural analysis of PC materials should be provided.

Response 2: Thanks for the suggestion. As suggested, we have now provided more structural analysis of PC substrate. We have characterized the PC using *ex-situ* infrared spectroscopy (please see **Fig. R2**). Moreover, we have offered more SEM images of the PC substrate (please see **Fig. R3**). As for the pores, we have employed both Brunauer-Emmett-Teller (BET) method (**Fig. R4a, b**) and mercury intrusion porosimetry (MIP) methods (**Fig. R4c, d**) to investigate the porous structure of PC since pores are highly related to the eNSR/eASR performance. PC shows a high BET surface area of 685.03 m² g⁻¹. The predominant pore diameter for PC is around 3.6 nm, suggesting the existence of numerous mesopores (2–50 nm) in PC. The MIP technique used in this study can only measure pores with diameters >5 nm. In **Fig. R4c**, the y-axis represents the cumulative mercury intake volume during the pressure increase. The material (*i.e.*, the PC or the NCP/PC) absorbs a large amount of energy during the feeding process; during withdrawal, as the pressure decreases, the material undergoes volume expansion due to stress relief, and the resulting fissures or void spaces that have changed location are filled with mercury, thus resulting in a withdrawal curve that is higher than the feed curve (please refer to: *Sci. Rep.* **10**, 22353 (2020)). The mercury intrusion pore size distribution curve of the PC (**Fig. R4d**) shows that PC exhibits precise macroporous structures (please refer to: *Chinese J. Catal.* **34**, 1534–1542, (2013), *Chem. Eng. J.* **163**, 389–394, (2010)); the macropores are centered at ~2.5 μm and ~13.7 μm and in the range from ~35.3 μm to ~66.2 μm. **Table R3** provides more details of the structural information of PC.

Fig. R2 Infrared signals of the PC in the range of 500–4000 cm⁻¹ obtained from Fourier transform infrared spectroscopy measurement. The result suggests stretching vibration of –OH (3434.15 cm⁻¹), bending vibration of absorbed water molecules (1634.79 cm⁻¹), H–O deformation vibration (1382.62 cm⁻¹), and existence of H–O (1046.89 cm⁻¹), *etc.*, in the PC sample (please refer to: *Energy Fuels* **35**, 18815–18823 (2021), *Compos. B Eng.* **224**, 109169 (2021)).

Fig. R3 More high-magnification SEM images of the PC.

Fig. R4 (a) Nitrogen adsorption-desorption isotherms and (b) the corresponding pore size distribution curve for the PC. (c) Mercury feeding and mercury withdrawal curves and (b) MIP pore size distribution curve for the PC.

Table R3. Pore structure data of 3D PC by both BET and MIP characterizations.

Data	PC
BET surface area	685.0335 m ² g ⁻¹
Single point adsorption total pore volume of pores	0.303044 cm ³ g ⁻¹ (pores less than 346.1816 nm diameter)
Single point adsorption total pore volume of pores less than 40.3122 nm diameter	0.300800 cm ³ g ⁻¹
Desorption average pore diameter	5.0962 nm
MIP porosity	50.1493%
Interstitial porosity	47.6300 %
Median Pore Diameter (Volume)	10347.0 nm
Median Pore Diameter (Area)	6.8 nm
Average Pore Diameter (4V/A)	610.2 nm

Comment 3: The authors demonstrated that the NCP/PC achieve the smallest average diameter and the most modest migration velocity. The authors should clarify this issue carefully by experiments and theoretical evidences.

Response 3: Thank you for your helpful advice. The bubble diameters and migration velocities provided in the manuscript were statistically calculated by a multiphase flow measurement analysis software (Revealer, <https://www.revealerhighspeed.com/>). In the original submission, the data presented were drawn from statistical data from computer software over a relatively short period of observation, while in the revised manuscript we have collected more data over a longer period of time to more accurately reflect the facts about bubble diameter and migration velocity. The observed results still indicate that the optimal sample, NCP/PC, has the smallest average bubble diameters as well as the most modest migration velocity (Fig. R5). We have performed both experiments and theoretical calculations to address your concern, and the data can be found in Figs. R6-R9. Moreover, we apologize that when we first submitted the manuscript, the unit of the bubble diameter was incorrectly labeled as nanometers (which should be micrometers), and we have made corrections. The results based on more data can be seen in Fig. R5.

Fig. R5 Statistical results of H₂ bubble behaviors based on more data. The second row shows the statistical results of the rise velocities of bubbles and the average bubble diameters for various eNSR cathodes (The majority of the bubbles that were counted were basically moving bubbles, not adsorbed on the electrode).

Moreover, we have provided *operando* bubble release videos of different NCP-based electrodes in the revised manuscript (please see **Supplementary videos 1-6**) in order to make readers better understand their differences in bubble release. The migration of bubbles released by the electrodes, including NCP/PC, NCP/NF, NCP/TM, NCP/CC, NCP/GF, and NCP/CP, can be directly observed through the videos. Compared to the other electrodes (please see **Supplementary videos 1-5**), the NCP/PC has the best bubble release behaviors (please see **Supplementary video 6**).

1) *For bubble diameters (experiments evidences):*

To understand why the NCP/PC has the smallest average bubble diameters, we have recorded the water contact angles and air contact angles of all the NCP-based samples (the corresponding revisions on air contact angles can be found in the revised **Fig.4k** or **Fig. R6**). The order of the samples with the air contact angle from large to small is: NCP/PC (151.12°) > NCP/NF (147.75°) > NCP/CP (147.17°) > NCP/CC (145.70°) > NCP/NF (147.75°) > NCP/TM (137.53°). With the water contact angle data in the revised **Fig.4j**, it thus conforms that the NCP/PC is the most hydrophilic and aerophobic electrode, which also means that bubbles can easily detach during the formation process. Therefore, we can observe that the NCP/PC has the smallest bubble sizes. Moreover, we have also performed adhesive forces measurements of gas bubbles on different NCP-based electrodes (the corresponding revisions can be found in revised **Fig.5a-f** or **Fig. R7**). The following trend indicates an increase in the bubble stretch forces (or the electrocatalyst-bubble interfacial adhesion force): NCP/PC (21.8 μN), NCP/NF (30.3 μN), NCP/CP (30.7 μN), NCP/CC (39.9 μN), NCP/GF (42.3 μN), NCP/TM (46.8 μN). The bubble-electrode adherence can lead to lower seawater reduction efficiency. The smallest adhesive force (21.8 μN) for NCP/PC thus demonstrates that it should exhibit faster bubble evolution kinetics.

Fig. R6 Bubble contact angle measurements under natural seawater of different NCP-based electrodes. The average contact angle for each electrode are derived from three separate tests. All data here were obtained using a fully automatic measuring device (Dataphysics OCA50, Germany).

Fig. R7 Gas bubble adhesive force measurements on various NCP-based electrodes. The inset photos shows the bubble behaviors in different processes. All data here were obtained using a high-sensitivity micro-electromechanical balance system (Dataphysics DCAT21, Germany).

(2) For bubble rise velocities:

Based on **Figs. R6** and **R7**, NCP/PC has the highest average contact angle and the smallest electrode-bubble interfacial adhesion force, making the bubbles on its surface

the easiest to escape. Therefore, among all the NCP-based electrodes, NCP/PC releases bubbles at the fastest speed. Moreover, bubble sizes on NCP/PC should thus be smaller than those on other electrodes, which has also been confirmed by *operando* microscope observations. According to Stokes's Law and/or Hadmard–Rybczynski's equation, a smaller bubble diameter leads to a slower bubble rising speed (please refer to: *Adv. Colloid Interface Sci.* **246**, 40-51, (2017)). Therefore, even though the NCP/PC releases bubbles with the fastest efficiency, rising speeds of the small bubbles are actually slower than those of relatively larger bubbles released by other NCP-based electrodes. This should be one of the reasons for most modest bubble migration velocity. Actually, the rise velocity results of our statistics is in line with expectations, because the proportions of bubble rising speed exceeding 0.04m/S for NCP/PC are smaller than those of bubble rising speed below 0.04m/S for NCP/PC.

(3) *For bubble diameters and rise velocities (from the perspective of theoretical calculations):*

In addition to newly added experimental data, we have performed simulations with finite element methods to have a deeper understanding of the most modest migration velocities for released bubbles on the surface of NCP/PC. As the bubble diameter decreases, the related bubble slip velocity ($u_{\text{gas}}-u_{\text{liquid}}$, the velocity of bubble phase relative to liquid phase) decreases, but the pushing forces on the liquid phase are strengthened. The liquid velocity sharply increases when bubble diameter approaches a particular point (the tipping point), which causes the bubble velocity to increase rather than decrease (**Fig. R8a** or **Supplementary Fig. 57a**). Simulation results reflect the trends of such changes (**Fig. R8b** or **Supplementary Fig. 57b**). Since the bubble diameters of NCP/PC are much smaller than those of all other NCP-based samples, bubbles released by NCP/PC will be closer to the tipping point, resulting in the smallest bubbles with moderate moving velocities. Bubbles are forced to attempt to cluster together by the gas-liquid interfacial tension to further reduce the overall surface area. This prevents bubbles from breaking free from the gas cluster. The detachment of individual bubbles (i.e., to break free from the gas cluster) thus requires a perturbation, which may originate from inhomogeneity in (1) pore geometry or (2) reaction surface

density for gas production inside the electrode. The inhomogeneity in the liquid inlet velocity (please see **Supplementary Note 6**) creates the perturbation. The gas-liquid interface of the optimal sample is more sensitive to the perturbation, presenting the obvious convex/protuberance (**Fig. R9a** or **Supplementary Fig. 58a**); however, the counterpart sample's pore connectivity is too strong (**Fig. R9a** or **Supplementary Fig. 58b**), making it harder for the perturbations to affect the gas-liquid interface and thus difficult to encourage detachment of bubbles. Moreover, the bubbles of the optimal sample coalesce only on the outer surface of the electrode, while the bubble of the counterpart sample has coalesced in it, which is another reason for the larger bubble of the counterpart sample.

Fig. R8 (a) A brief schematic diagram of velocity magnitude versus bubble diameter. (b) Relationship between the stabilized velocity of the liquid phase (u_{gas} , time-averaged from $t = 30$ s to 50 s) and the bubble diameter ($i_0 = 0.125$ A).

Fig. R9 Comparison of the volume fraction of gas, the ϕ_g , for (a) the optimal electrode and (b) the counterpart electrode.

Comment 4: As shown in Figure 3f, the authors emphasize that the massive insoluble precipitates formed and accumulated on the NCP/NF after the 10-h eNSR testing, but these pictures provided in the manuscript are difficult to support each component and structure. The author needs to clarify this issue carefully by in-situ evidences.

Response 4: Thanks very much for this comment. As suggested, we have performed *in-situ* Raman spectroscopic measurements for all NCP-based electrodes (including the NCP/PC) to further characterize the surface insoluble precipitates. The corresponding results can be found in **Fig. R10**, **Fig. R11**, and **Fig. R12**. As shown in **Fig. R10**, for the period of time at the beginning of natural seawater reduction electrolysis, Raman spectra of most electrodes exhibit peaks at around 450 cm^{-1} , which should be attributed to metal phosphides (please refer to: *Nat. Commun.* **14**, 1997, (2023), *Chem. Eng. J.* **473**, 145397, (2023)). However, with the extension of electrolysis time, most electrodes (except for NCP/PC) lose the peak signals of M–P during electrolysis, which implies that there is no metal phosphide on their surfaces. In contrast, although the peak density of M–P for NCP/PC also decreases, the corresponding Raman peaks can still be

observed after 60 min of electrolysis, indicating the strong anti-precipitation ability of NCP/PC. As shown in **Fig. R10b**, The peak at around 669 cm^{-1} can be observed for NCP/NF, corresponding to Eg (R) mode of $\text{Ca}(\text{OH})_2$ (please refer to: *Solid State Commun.* **8**, 541–543 (1970)). This peak belonging to $\text{Ca}(\text{OH})_2$ can also be observed in spectra for several other electrodes, including NCP/CC, NCP/GF and NCP/CP. This shows that $\text{Ca}(\text{OH})_2$ is formed and accumulated on the surfaces of these electrodes. Moreover, the cathodic electrodes, including NCP/NF (**Fig. R10b**), NCP/TM (**Fig. R10c**), NCP/CC (**Fig. R10d**), NCP/GF (**Fig. R10e**), and NCP/CP (**Fig. R10f**), all shows visible Raman bands in between $\sim 1000\text{ cm}^{-1}$ and $\sim 1300\text{ cm}^{-1}$. The Raman bands in this range (~ 1000 to $\sim 1300\text{ cm}^{-1}$) further indicate that the surface precipitates should contain CaCO_3 and $\text{Ca}(\text{OH})_2$ (please refer to: *RSC Adv.* **6**, 104537–104548 (2016), *Phys. Chem. Minerals* **46**, 229–244 (2019), *Acta Geotech.* **16**, 3229–3237 (2021). *Chem. Commun.* **53**, 6657–6660 (2017), *Appl. Environ. Microbiol.* **81**, 7403–7410 (2015), *Heritage* **4**, 3970–3986 (2021), *J. Appl. Electrochem.* **39**, 39–44 (2009), *J. Power Sources* **172**, 435–445 (2007)). Noticeably, the Raman data for NCP/PC (**Fig. R10a**) show no obvious band extending from ~ 1000 to $\sim 1300\text{ cm}^{-1}$, again confirming the strong anti-precipitation ability of NCP/PC. In addition, *in situ* Raman spectra in higher wavenumbers ranging from 2000 cm^{-1} to 4000 cm^{-1} in **Fig. R11** reveal that NCP/NF (**Fig. R11b**), NCP/TM (**Fig. R11c**), NCP/CC (**Fig. R11d**), NCP/GF (**Fig. R11e**), and NCP/CP (**Fig. R11f**) are all covered by insoluble $\text{Mg}(\text{OH})_2$ precipitates (please refer to: *Sci. Rep.* **6**, 20525 (2016), *J. Geophys. Res. Solid Earth* **124**, 8267–8280 (2019)). In sharp contrast, NCP/PC (**Fig. R11a**) does not show any Raman peak in this wavenumber range, fully indicating its superb anti-precipitation ability. Except for the time-dependent data in **Fig. R10** and **Fig. R11**, potential-dependent Raman analysis in **Fig. R12** again reveals little precipitation on NCP/PC under various applied potentials.

Fig. R10. *In-situ* electrochemical Raman spectra for various NCP-based cathodes in unprocessed natural seawater (wavenumber range: from 100 cm^{-1} to 2000 cm^{-1}), including the (a) NCP/PC, (b) NCP/NF, (c) NCP/TM, (d) NCP/CC, (e) NCP/GF, and (f) NCP/CP. The time-dependent data here were recorded under a fixed electrode potential.

Fig. R11. *In-situ* electrochemical Raman spectra for the different NCP-based electrodes in unprocessed natural seawater (wavenumber range: from 2000 cm^{-1} to 4000 cm^{-1}), including the (a) NCP/PC, (b) NCP/NF, (c) NCP/TM, (d) NCP/CC, (e) NCP/GF, and (f) NCP/CP. The time-dependent data here were recorded under a fixed electrode potential.

Fig. R12. *In-situ* electrochemical Raman spectra for NCP/PC under different electrode potentials in natural seawater.

Comment 5: *In order to better clarify the anti-precipitation advantage of the PC structure, a comparison of the ECSA changes of NCP with different substrates (CP, CC, GF, TM, and NF) after 10 h electrolysis should be provided.*

Response 5: Thanks very much for raising this critical concern. As suggested, we have provided a comparison of the ECSA changes of NCP with different substrates (PC, CP, CC, GF, TM, and NF) after 10 h of continuous electrolysis (please see **Fig. R13** and **Fig. R14**). Electrochemical C_{dl} values for NCP with different substrates (PC, CP, CC, GF, TM, and NF) after the 10 h of NSR electrolysis are 83.1 mF cm^{-2} for the NCP/PC, 4.3 mF cm^{-2} for the NCP/CP, 1.7 mF cm^{-2} for the NCP/CC, 5.4 mF cm^{-2} for the NCP/GF, 4.0 mF cm^{-2} for the NCP/TM, and 4.8 mF cm^{-2} for the NCP/NF. Changes in the electrochemical C_{dl} value (ΔC_{dl}) for different electrodes after the 10 h of NSR electrolysis are 3 mF cm^{-2} for the NCP/PC, 26.1 mF cm^{-2} for the NCP/CP, 33.8 mF cm^{-2} for the NCP/CC, 49.8 mF cm^{-2} for the NCP/GF, 64.6 mF cm^{-2} for the NCP/TM, and 67.3 mF cm^{-2} for the NCP/NF. The NCP/PC shows the smallest ΔC_{dl} , which is in line with minimal surface precipitation.

Fig. R13 Electrochemical C_{dl} measurements for different anodes after 10 h of NSR electrolysis. Cyclic voltammetry curves at various scan rates ranging from 10 to 50 mV s^{-1} within the non-Faradaic potential range for (a) NCP/PC, (b) NCP/NF, (c) NCP/TM, (d) NCP/GF, (e) NCP/CC, and (f) NCP/CP.

Fig. R14 Capacitive current densities of voltammetry curves as the function of scan rates correspondingly. The C_{dl} for the reconstructed NCP/PC is 83.1 mF cm^{-2} , close to that of pristine NCP/PC (86.1 mF cm^{-2}). Therefore, only the NCP/PC demonstrates a minimal change in ECSA. The remaining NCP-based electrodes all have drastically decreased ECSAs, which should be resulted from the heavy precipitation covering the catalytically active sites.

Comment 6: *Electronic metal–support interactions (EMSI) modulation mechanism of NCP on various substrates should be given. The authors may need to describe the reaction mechanism in more details. Discuss clearly importance of NCP and correlation to the substrates.*

Response 6: Thank you very much for your comment. We do understand your concern. First of all, the interaction between metallic center and basement was firstly figured out by Tauster in 1978, who put forward the term of the metal–support interaction (MSI) (please refer to: *J. Am. Chem. Soc.* **100**, 170–175 (1978)). In 2012, the term “electronic metal–support interaction” (EMSI) was first coined by C. T. Campbell et al. based on the interaction between platinum and ceria atoms. Generally, EMSI can be divided into

three types, metal–metal substrate, metal–transition metal compounds (M–TMCs), and metal–doped carbon (M–dN) (please refer to: *Adv. Mater.* **32**, 2003300 (2020)). EMSI is thus more often used for describing interactions between metals (especially noble metals) and nanoscale supports. Please note that our substrates (such as PC, GF, and CC, *etc.*, in this work) are not nano-sized supports.

In addition, according to our control experiments, the performance of NCP/PC is far superior to other electrodes, not because of the electronic metal–support interactions between NCP and bulk substrates, but because of the unique 3D structure of the honeycomb-type PC. For instance, similar to NCP/PC, the related counterpart electrodes, NCP/CC, NCP/GF, and NCP/CP, also contain the carbon substrates; however, the performance of NCP supported by these carbon structures (*i.e.*, CC, GF, and CP) is not as good as that of NCP/PC, especially the anti-precipitation ability. Thus, the significant increase in performance of NCP/PC (compared with other NCP-based electrodes) is due to the structural advantages of honeycomb-type PC.

Comment 7: A series of in situ and ex situ characterizations to investigate the effect of transport/traffic micro-systems on the structural stability of NCP on various substrates for HER in seawater systems. Meanwhile, the author may consider establish the approximate linear relationship of Mg^{2+}/Ca^{2+} precipitates and efficient traffic of bubbles.

Response 7: We think this is a good suggestion. We have made both experimental and theoretical attempts to explore such relationships. First of all, based on your helpful suggestion on establishing an approximate linear relationship of Mg^{2+}/Ca^{2+} precipitates and efficient traffic of bubbles, we have performed a series of experiments. The theoretical part is detailed in **Response 9**. For the experiments, we immersed NCP/PC in an aqueous solution with the same concentration of agarose gel at varied depths (a same soaking time) in order to block the pore structure of NCP/PC to varying degrees. To be more specific, NCP/PC being immersed 1/4 of its volume will lose 25% of the pore structure (75% of the available pores are thus left, denoted as NCP/PC-75%), NCP/PC being immersed 1/2 of its volume will lose 50% of the pore structure (denoted

as NCP/PC-50%), NCP/PC being immersed 3/4 of its volume will lose 75% of the pore structure (denoted as NCP/PC-25%), and NCP/PC being fully immersed in agarose gel-containing solution will lose 100% of the pore structure (denoted as NCP/PC-0%). Meanwhile, two more samples with extended soaking times include NCP/PC-3 min and NCP/PC-10 min.

As shown in **Fig. R15** and **Fig. R16a**, the more parts of the NCP/PC electrode that are immersed in the aqueous solution of agarose gel, the more pores are blocked, *i.e.*, the more damage is done to the pore structure (the smaller the ECSA), the lower the j will be and the more $\text{Mg}^{2+}/\text{Ca}^{2+}$ precipitates should accumulate. Conversely, the less parts of the NCP/PC electrode that are immersed in the aqueous solution, the less pores are blocked, *i.e.*, the less damage is done to the pore structure (the higher the ECSA), the higher the j will be and the less $\text{Mg}^{2+}/\text{Ca}^{2+}$ precipitates will accumulate.

As shown in **Fig. R16b**, the more parts of NCP/PC that are immersed, the lower j of the electrode, which indicates a decrease in bubble release. Besides, the more NCP/PC is immersed, the smaller the corresponding ECSA is (**Fig. R16a**), validating more destroyed pore structure as well as less surface area for electrocatalysis reaction. The observation thus shows that the transport of bubbles is hindered by the introduction of agarose gel. We obtained the relationship between amounts of $\text{Mg}^{2+}/\text{Ca}^{2+}$ precipitates on the electrode and the related j (*i.e.*, the degree that represents the destruction of the pore structure) in **Fig. R17**. The data do show a near linear relationship that the greater the j (*i.e.*, the less destroyed pore structure as well as the more efficient bubble release), the less $\text{Mg}^{2+}/\text{Ca}^{2+}$ precipitates.

In addition, we have performed detailed finite element calculations to theoretically and quantitatively compare the abilities of the electrodes to repel precipitates under different bubble traffic conditions, *i.e.*, optimal sample with efficient traffic of bubbles and counterpart electrode with less efficient traffic of bubbles. Please see (1) the detailed quantificational comparison of the repelling ratio ($\eta(t)$) between the optimal electrode and the counterpart electrode as well as (2) the relationship between the η (efficiency in repelling precipitates) and the gas production volume per unit area (*i.e.*, the traffic of the bubbles) for the optimal electrode in **Response 9**.

Fig. R15 Electrochemical C_{dl} measurement results for NCP/PC with different pore blocking treatments.

Fig. R16 (a) C_{dl} values and (b) the corresponding changes in eNSR j versus the applied potentials for NCP/PC with different pore blocking treatments.

Fig. R17 Linear fitting of j and precipitation amounts for NCP/PC with different pore blocking treatments. Note that j here indirectly represent whether the bubble release is effective or not.

Comment 8: The authors claim that NCP/PC holds better HER performance compared to other PC electrodes (CoP/PC, Ni₂P/PC, MoS₂/PC), which may contribute to the optimized anti-precipitation ability, while a comparative analysis of their performance as well as single NCP, CoP, Ni₂P and MoS₂ should be provided.

Response 8: As suggested, we have provided a comprehensive comparative analysis of the performances of all the samples in revised **Supplementary Figs. 46** and **50** as well as **Figs. R18** and **R19**, including single NCP, CoP, Ni₂P and MoS₂ supported by CP.

Fig. R18 (a) LSV comparison, (b) Tafel slopes, (c) C_{dl} values, and (d) Nyquist plots of different PC-based electrodes in natural seawater, including NCP/PC, Ni₂P/PC, CoP/PC, and MoS₂/PC. Polarization curves of PC and Pt/C/PC and Tafel slope of Pt/C/PC are also given.

Fig. R19 Comparison of eNSR activities of single NCP, CoP, Ni₂P and MoS₂ supported by CP. (a) Linear sweep voltammetry curves, (b) Tafel slopes, (c) C_{dl} values, and (d) Nyquist plots of different PC-based electrodes in natural seawater, including NCP/CP, Ni₂P/CP, CoP/CP, and MoS₂/CP.

Comment 9: The real-time high-definition images provided seem to be difficult to obtain direct evidence about the microscopic bubble/precipitate traffic system (MBPTS), which is also the main highlight of the manuscript, so clearer hints and key evidence need to be given.

Response 9: We thank the reviewer for this very helpful comment. As suggested, apart from added experimental data in revised Manuscript, revised Supporting Information, and theoretical simulation results in **Figs. R8** and **R9**, we have performed more finite element simulations involving two models of NCP/PC (i.e., the optimal sample) and the counterpart electrode in order to provide more in-depth and comprehensive insights on the MBPTS. We defined the repelling ratio to precipitates as $\eta(t)$. The detailed methods of all theoretical simulating can be found in **Supplementary Notes 5** and **6**

and **Supplementary Figs. 55 and 56**. Notably, **Figs. R20** shows the distribution results of precipitation concentration during the reaction. Due to a more directional movement of bubble flows in the optimal sample, the precipitates can be discharged efficiently (**Figs. R20a** or **Supplementary Fig. 59a**); however, in the counterpart sample, due to the non-directional flow of the electrolyte solution (i.e., more random directions, see **Figs. R20b** or **Supplementary Fig. 59b**), the precipitate discharge ability is not as good as that of the optimal sample. Furthermore, **Fig. R21** shows that vortices are easy to appear in the velocity field, this consists with the lower precipitate discharge ability stemmed from non-directional flows in the counterpart sample. We also quantified the amount of precipitates diffusing out of the pores for both two models and compared the theoretical repelling ratios at different times. **Fig. R22** compares the repelling ratios of the two electrodes, with lower $\eta(t)$ for the counterpart sample. Moreover, **Fig. R22b** shows the data from the relationship between the η and the gas production volume per unit area, which is the theoretical results for responding your comment 7. In addition, we have tried to optimize the pore sizes of the electrode theoretically, and the corresponding results are summarized **Supplementary Figs. 62–65**.

Fig. R20 Visualized comparison of distribution of the precipitation concentration. (a) Precipitation concentration distribution of the optimal electrode that represents

NCP/PC and (b) the counterpart electrode.

Fig. R21 Vortices that are prone to appear in the velocity field. This non-directional flow field is not conducive to precipitation repulsion.

Fig. R22 (a) Quantitative comparison of the optimal electrode and the counterpart electrode in terms of the $\eta(t)$. (b) The relationship between the η (efficiency in repelling precipitates) and the gas production volume per unit area (i.e., the traffic of the bubbles) for the optimal electrode.

In addition to the theoretical explanations presented by finite element simulations, we have provided other possible reasons for the effective operation of the microscopic bubble/precipitate traffic system of our NCP/PC to further answer to your valuable question. The detailed discussions can also be found below or in **Supplementary note 3**.

(1) Honeycomb-like structure of NCP/PC with hydrophilic, open, well-arranged, low-tortuosity micro-channels

Unlike NF and GF with disordered pore structures, PC has an ordered honeycomb-like porous structure that allows bubbles to escape more quickly without being hindered during the moving process (please refer to: *Adv. Energy Mater.* **10**, 2002955 (2020)). During the eNSR, the open, well-arranged, low-tortuosity micro-channels of NPC/PC are perfect for the quick transportation of electrons, intermediates, gases, *etc.* Such aligned and hydrophilic channels within the NCP/PC provide extra vent pathways for electrolytes permeation and gaseous bubble escape, thus facilitating the mass transfer. Moreover, NPC grown almost vertically on PC walls but maintained the PC framework unimpeded, which is believed to facilitate electrolyte diffusion and expedite gas escape. In contrast, bubbles on other NCP-based electrodes stayed for a long time and gathered into a big size before leaving the surface, obstructing the active sites and preventing the electrolyte from diffusing.

(2) Larger specific surface areas of the smaller bubbles contribute to better abilities to repel precipitates

The smaller the sizes of the gas bubbles, the larger specific surface areas of the bubbles, and the more precipitates the bubbles would carry away, hence the best anti-precipitation capacity of NCP/PC.

(3) Hierarchically porous structure of NCP/PC

Benefiting from the low-tortuosity and hierarchically porous structure, the electrolyte can penetrate into pores of the NCP/PC, and the numerous gas bubbles generated on the catalyst surface is favored to be released from the hierarchically porous structures (especially the directional channels), thereby ensuring the pathways of mass transfer. Notably, apart from the wood channels, mesopores (2–50 nm) can also help H₂ gas

release and electrolyte permeation during HER process (please refer to: *Electrochim. Acta* **330**, 135274 (2020)). Furthermore, holes across the wood-derived electrode in the perpendicular direction can connect the parallel/aligned channels, thereby enabling enable better electrolyte diffusion/ion exchange between the channels and reducing flow resistance/concentration polarization (please refer to: *Energy Storage Mater.* **27**, 327–332 (2020)).

(4) Inner burst forces

As the seawater reduction reaction proceeds, H₂ bubbles in the internal low-tortuosity channels of NCP/PC gradually become more numerous and merge to increase in size, and soon the entire bubble diameter that forms may reach the diameters of the channels (see Fig. R23a–c). Due to space restriction effect of channels, H₂ bubbles would burst as soon as the diameters approach those of the channels of NCP/PC, generating burst forces (see Fig. R23d). In the interface, the acquired blasting force disperses numerous OH⁻ (i.e., Ca/Mg precipitates) away from catalytic centers, and thus efficiently avoids the poisoning of active sites (please refer to: *Electrochim. Acta* **330**, 135274 (2020)). Therefore, burst forces derived from the breakage of bubbles within the channeled structure of NCP/PC may contribute to the operation of the bubble/precipitate traffic system.

Fig. R23 Schematic diagram of the cause of the burst forces. Gas bubbles grow inside the NCP/PC electrode and eventually burst.

(5) Small stretch forces & Buffer of stress/tension or volume expansion caused by bubble desorption

According to a study by Liu et al. (please refer to: *Adv. Funct. Mater.* **32**, 2107308,

(2022)), internal hollow cavities of NCP/PC should **buffer the stress/tension or volume expansion caused by bubble desorption**. Except for this reason, smaller bubbles produce smaller stretch forces (please refer to: *Adv. Funct. Mater.* **32**, 2107308, (2022)). For the NCP/PC, **the smallest stretch force** induced by the smallest bubbles will minimize the damage of the electrode. Thus, both the two reasons guarantee the longer lifespan of NCP/PC during seawater oxidation process.

(6) The least amount of inactive areas

Since NCP/PC exhibits the best seawater reduction performance, the nucleation and growth of bubbles on its surface would result in **the least amount of inactive areas** separating the seawater from the active sites (please refer to: *Chinese Chem. Lett.* **35**, 108351, (2024), *J. Am. Chem. Soc.* **142**, 1857–1863, (2020), *Appl. Catal. B* **286**, 119920 (2021)). Bubbles that grow on the electrode over a long period of time can contribute to more inactive areas separating the seawater and the catalytic surface. The smaller bubble diameter means lower gas-phase occupancy and higher exposure rate of active areas, which indicates higher HER activity (please refer to: *Appl. Energy*, **307**, 118278 (2022)).

(7) Less gas transfer resistance

In situ microscopic observation was performed to directly showcase bubble evolution behaviors for different NCP-based anodes at the same electrolysis voltage. The large numbers of generated gas bubbles of NCP/PC may result from the higher ECSA that generates more catalytic sites. Moreover, the larger quantity of bubbles and the smaller bubble sizes for our NCP/PC indicate **less gas transfer resistance** for the NCP/PC (please refer to: *Nano Lett.* **23**, 629–636 (2023)), which facilitates the operation of microscopic bubble/precipitate traffic system. According to the research on the relationship between bubble release and local electrolyte potential, large bubbles will greatly disturb the local electrolyte potential and seriously hinder the subsequent bubble generation (please refer to: *Nano Lett.* **23**, 629–636 (2023)). Therefore, the small bubbles of NCP/PC make the **local electrolyte potential less affected**, leading to increased energy utilization efficiency (please refer to: *Nano Lett.* **23**, 629–636 (2023)). Additionally, bubbles with smaller sizes are better at facilitating charge transfer and

solution resistance (please refer to: *Adv. Funct. Mater.* **32**, 2107308, (2022)), which lowers the potential during gas-evolving electrocatalytic reactions.

In conclusion, using just low-cost natural pine wood blocks as the electrode precursor, it is now possible to create self-cleaning eNSR electrodes. Importantly, smallest bubbles, more directional movements of bubble flows, fewer vortices, higher sensitivity of gas-liquid interface to perturbation, the least amount of inactive/dead areas, less gas transfer resistance, and less affected local electrolyte potential, etc., may all allow PC-based electrodes like NCP/PC to achieve the more effective microscopic bubble/precipitate transport system.

Comment 10: *Optical photographs of the Faradaic efficiency test also need to be provided.*

Response 10: As suggested, we have provided the optical photographs of the FE tests in revised **Supplementary Fig. 67** as well as **Fig. R24**.

Fig. R24. Optical photographs of the drainage tests and the related produced gas amounts for Faradaic efficiency calculations. We conducted the gas quantification tests at regular intervals and determined the each FE value in the selected range (25-min testing for each measurement). The H₂ gas produced from the cathodic chamber will be fed into the drainage tube.

Point-by-point response to the reviewers #3

Reviewer #3 (Remarks to the Author): *Sun and co-workers described an emerging 3D micro/nano-structured electrodes that combine merits of honeycomb-like carbon skeleton and H₂-evolution nanocatalyst for eASR/eNSR. Bubbles released by NCP/PC achieve the most modest migration velocity and the smallest average diameter. It also operates at the ampere-level current density of -1 A cm^{-2} for at least 1000 h without failure. The flow-type electrolyzer with NCP/PC||DSA stably functions at 500 mA cm^{-2} for 150 h without any discernible attenuation, while sustaining high H₂ FE values (above 96.5%) throughout the 150 h. This is a routine addition to a well-known field. This manuscript suffers from following lacks:*

General Response: We sincerely appreciate the reviewer for the insightful comments on our work. We acknowledge that your comments are important for us to improve the quality of our work. We have studied all comments carefully and have made corrections which we sincerely hope meet with approval.

Comment 1: I noticed the size of electrode is quite small, and the thickness of the electrode is very large. Therefore, it is very easy to obtain very large current density (e.g. 500 mA cm^{-2}) in lab. In an industrial electrolyzer, the electrode is very large and the distance between cathode/anode is very small. The foam is applied in most cases, but its thickness is very small. Therefore, the validation of very large current density at an industrial electrolyzer is missing now.

Response 1: Thanks for raising the thought-provoking comments. First of all, in the initial submitted manuscript, the electrode sizes we tested were already consistent with, and even larger than, many of the recent seawater splitting studies (please refer to: 0.5 cm^2 for *Energy Environ. Sci.*, 2022, **15**, 3945–3957, 0.126 cm^2 for *Adv. Funct. Mater.* **32**, 2201127, (2022)). Besides, we point out that the size of the wood precursor will be significantly reduced after long-time carbonization (a 6-h carbonization process under $1000 \text{ }^\circ\text{C}$ in this work), hydrothermal and phosphidation treatments, generally to a shrunken thickness of less than 2 mm. In fact, the thickness of our NCP/PC is even

lower than the commercial Ni foam (please see **Fig. R25**). Moreover, the carbonized wood (i.e., the PC) costs less than many commercial substrates. The prices for the commercial substrates (Ni foam, Cu foam, NiFe foam, carbon paper, etc.) can be found on Amazon.com, on Taobao.com, and in local chemical companies. Last but not the least, we must point out that, as can be seen in **Fig. 3** of the manuscript, the performance of NCP/PC far exceeds that of the rest of the NCP catalysts on various substrates, including the Tafel slope, mass activity, anti-precipitation ability, *etc.* Note that such capabilities (e.g., the anti-precipitation ability) can hardly be improved by changing the geometric area of the electrode.

We have collected more experimental data to further address your concern. We have prepared NCP/PC electrodes with different sizes for electrochemical seawater reduction tests (in a three-electrode system), and the results are less different from the current densities in the initial submission (please see **Fig. R26**). Be aware that most reports in the field of seawater electrolysis do not use electrodes larger than 1 cm². Our prepared electrodes still exhibit better activities even at the size of 4cm² as well as 9cm², and the electrolyte is natural seawater.

Fig. R25 Photos of classical Ni foam and our NCP/PC. (a) Photos of our NCP/PC. Inset shows the photo taken from another angle. (b) Photos of a commercial Ni foam used for the electrolysis test. (c) Electrodes held together by a pair of tweezers to better visualize the thick difference. The photos here illustrate that the NCP/PC is not thick, and the thickness comparison of the two electrodes shows the thinner thickness of our NCP/PC.

Fig. R26 Polarization curves recorded with NCP/PC electrodes in different sizes (0.25 cm², 1 cm², 4cm², and 9cm²). The measured electrochemistry data here were obtained in a three-electrode system with natural seawater as the electrolyte.

There are two main reasons why industrial water electrolyzers (IWEs) have not been used in our work. First, little research has been done on seawater electrolysis using the conventional IWEs. This is because conventional IWEs are unsuitable for the long-term operation in seawater environments due to the high complexity of seawater, especially natural seawater (please refer to: *Mater. Today* **69**, 193–235 (2023), *Sci. Adv.* **9**, eadi7755 (2023)). The improvement in electrolytic devices and cell components for seawater electrolysis is thus another crucial research area right now (please refer to: *Nature* **612**, 673–678 (2022), *Nat. Commun.* **14**, 3934 (2023), *Joule* **7**, 765–781, 2023, *ACS Energy Lett.* **8**, 2387–2394 (2023)). Secondly, some efforts to design high-performance electrodes have used industrially relevant membrane electrode assembly (MEA) electrolyzers to test whether the electrodes are suitable for industrial applications (please refer to: *Nat. Energy* **8**, 264–272 (2023)). Considering such latest

situations, in the first submission, instead of using conventional IWEs for nature seawater electrolysis, we assembled a flow-type electrolysis cell more suitable for natural seawater electrolysis for the demonstration. Many small precipitates can be discharged from our electrolytic device in time. This may be much more helpful for future research.

To better address your concern, we have further demonstrated seawater reduction activities of NCP/PC in a customized scaled-up electrolyzer as well, which has higher electrode surface areas than that reported in a recent work by Guo et al. (please refer to: *Nat. Energy* **8**, 264–272 (2023)) on nature seawater electrolysis (please see the photos of our customized electrolyzer consisting of single stacks in **Fig. R27a–c**). The scaled-up electrolyzer consists of the outer frame (10 cm × 10 cm × 9 cm) and independent cells (i.e., electrolysis stacks). Commercial ruthenium oxides (loaded on porous Ti felt, RuO₂/TP) are used for the anodes. The resulting current densities are still high (please see **Fig. R27b**), easily attaining 500 mA cm⁻² at the cell voltage of ~3.5 V with no *iR* compensation. Notably, the size of NCP/PC electrodes tested in our scaled-up electrolyzer (27 cm²) even exceeds most of the recent reports on alkaline seawater electrolysis (please refer to: an electrode size of 0.25 cm² for *Adv. Mater.* **33**, 2007508 (2021), an electrode size of <0.5 cm² for *Adv. Mater.* **33**, 2003846 (2021), the electrode sizes of 0.5 ~ 1 cm⁻² for *Adv. Mater.* **34**, 2201774 (2022), an electrode size of 4 cm² for *Adv. Mater.* **35**, 2210057 (2023), an electrode size of 1 cm² for *Angew. Chem. Int. Ed.* **62**, e202309854 (2023), an electrode size of 1 cm² for *J. Am. Chem. Soc.* **144**, 9254–9263 (2022), an electrode size of <1 cm² for *Energy Environ. Sci.* **13**, 3439–3446 (2020), the electrode sizes of 0.3 ~ 0.45 cm⁻² for *Nat. Commun.* **10**, 5106 (2019), an electrode size of 0.126 cm² for *Adv. Funct. Mater.* **32**, 2201127 (2022)), which fully demonstrates the good potential of our NCP/PC in seawater electrolysis-related applications.

Fig. R27 Performance measurements of the NCP/PC electrode in the electrolyzer consisting of single stacks. (a–c) Pictures of the DC power as well as the outside and inside of the scaled-up electrolyzer for industrially relevant testing. The total electrode area is 54 cm². (d) Polarization curve of the electrolyzer by measuring the voltage at a series of current densities (without iR compensation). (e) Long-term durability data. *Notes:* TF for RuO₂/TF stands for porous Ti felt.

Comment 2: *The actual seawater should be applied and demonstrated if this work is claimed as ultrastable seawater reduction electrocatalysis.*

Response 2: We understand your concern. We first show alkaline seawater performance of our electrode in **Fig. 2** in that a great number previously reported catalysts have been evaluated in alkalized seawater. Compared to earlier cathodes, our NCP/PC electrode is considerably superior in alkaline seawater reduction.

We emphasize that precipitation-less natural seawater electrolysis is the central focus of our research. We collected natural seawater from Huangdao District, Qingdao City, Shandong Province, China, and all of the tests in **Fig. 3**, **Fig. 4** (except for water contact angle test) and **Fig. 6** of our manuscript used raw, natural seawater (*i.e.*, the actual seawater). This is one of the great strengths of our work, which is the realization of precipitation-less real natural seawater reduction electrolysis. Since the seawater itself

is clean enough (no notable suspended/floating particles; please see the picture of our collected seawater in **Supplementary Fig. 72** as well as ion contents of the seawater in **Supplementary Table 8**), we used the seawater directly for electrolysis demonstration without filtering it. Note that previous natural seawater electrolysis work with state-of-the-art catalytic electrode (please refer to: *Nat. Energy* **8**, 264–272 (2023)) used filtered natural seawater, while our work did not filter the natural seawater, allowing for genuine direct natural seawater reduction demonstration. This is indeed a breakthrough in the field of natural seawater electrolysis (please refer to: *Mater. Today*, **69**, 193–235 (2023), *Nat. Energy* **8**, 264–272 (2023), *Sci. Adv.* **9**, eadi7755 (2023)).

In fact, the electrochemical stability of our seawater electrolyzers (please see **Fig. 6**) stands out from the reported two-electrode natural seawater electrolyzers (please see **Supplementary Table 7**). As far as we know, this is one of the longest electrolysis lifetimes of cathodic electrode in real natural seawater.

Comment 3: How to regulate the bubble size herein? The match between the pore size and operating current density is quite important.

Response 3: You've asked a really good question. One research direction in the field of water electrolysis is bubble size management. Regarding controlling bubble size, a lot of work has been made in the field. For instance, Wang's group prepared wettable and non-wettable porous electrodes to demonstrate that wettability regulation should be an effective strategy for bubble size regulation (please refer to: *Joule*, **5**, 887–900, (2021)). In their work, varied polytetrafluoroethylene coverage on the electrode surface enables different bubble sizes. Moreover, Sun's group demonstrated the importance of “superaerophobic” nanoarray electrode (rather than flat electrode) for decreasing the bubble detachment size about 10 years ago (please refer to: *Adv. Mater.* **26**, 2683–2687 (2014)).

In other words, the more hydrophobic or wettable the electrode is, the smaller the bubbles it releases. In the revised manuscript, we have added more related experimental data by performing bubble contact angle measurements (Dataphysics OCA50, Germany, please see **Fig.4**) and bubble adhesive force measurements (Dataphysics DCAT21,

Germany, please see **Fig.5**). Based on our data, the NCP/PC electrode, as the optimal NCP-based electrode, is superhydrophilic and superaerophobic, which facilitates the release of smaller bubbles. Moreover, NCP/PC shows the lowest electrocatalyst-bubble interfacial adhesion force among all NCP-based electrodes. Thus, the 3D honeycomb-like NCP/PC is naturally able to release smaller bubbles than other NCP-based electrodes by virtue of its own geometric properties.

Moreover, while the match between the pore size and operating current density has been extensively studied as an important research topic (please refer to: *J. Power Sources* **580**, 233380 (2023), *Adv. Energy Mater.* **10**, 2002955 (2020)), it is not the focus of our work. Our work reveals universal principles underlying superb anti-precipitation performances (*i.e.*, the honeycomb-like PC-based electrode design) to solve the long-standing issues in electroreduction of low-grade natural seawater.

Instead of regulating bubble size or matching pore size and current density, one primary finding of our research is that honeycomb-like PC-based electrode architecture, as a biomass-derived material, effectively realizes self-cleaning electrolysis in natural seawater. What is incredible about NCP/PC is that it naturally provides a special bubble release behavior to endow itself a powerful precipitation-repelling capacity without the need for artificial bubble/pore size regulation.

Comment 4: The evolution of catalyst surface functional groups should be demonstrated. It is widely accepted that the surface of many electrocatalysts has been oxidized/reduced into active states.

Response 4: Thanks for your kind reminders. We have employed a combination of *ex situ/in situ* techniques, including potential- or time-dependent Raman spectroscopy, X-ray photoelectron spectroscopy (XPS), and infrared spectroscopy (**Figs. R28–R31**), to comprehend the evolution of catalyst surface and active states. According to previous Raman studies on metal phosphides and NiCo-based materials during/after various electrochemical processes, such as HER catalysis, (please refer to: *Energy Environ. Sci.* **15**, 727–739 (2022), *ACS Catal.* **10**, 81–92 (2020), *Chem. Eng. J.* **473**, 145397 (2023), *Adv. Mater.* **34**, 2107548 (2022), *Int. J. Energy Res.* **46**, 13035-13043 (2022), *Nano*

Energy **114**, 108601 (2023), *Nat. Commun.* **14**, 1949 (2023), *J. Mater. Chem. A*, **9**, 18421–18430 (2021), *Appl. Catal. B* **316**, 121678 (2022), *Nat. Catal.* **4**, 1050–1058 (2021), *Adv. Energy Mater.* **13**, 2204114, (2023), *Chem. Eng. J.* **435**, 134261, (2022)), the two pronounced peaks appearing at the potentials more negative than -1.5 V in the Raman spectra of NCP/PC (**Fig. R28**) should belong to M–O bonds (M = Ni or Co) in metal oxyhydroxides.

We can make a simple determination of surface chemical compositions of electrode samples with the help of the full-scan spectra. Since the strongest peaks for almost all elements are located between 0~1200 eV, we provided the data of the sample before and after the electrochemical tests (**Fig. R29**). The XPS survey spectra show that the presence of Ni, Co, P and O elements in NCP/PC and NCP/PC after 150 h of eNSR (denoted as NCP/PC-150-used). For NCP/PC-150-used, the existence of Na element (~ 497 eV for Na Auger, ~ 1072 eV for Na 1s) can be attributed to the adsorbed Na species (from seawater) on surface. The existence of K elements (~ 377 eV for K 2s) can be attributed to the adsorbed K species (from seawater) on surface. The signal of P 2p is almost lost after the seawater reduction under eNSR conditions, which is consistent with the higher amount of phosphorus loss (compared to those of Ni and Co) according to the ICP-AES data. High-resolution XPS results (**Fig. R30**) and infrared data (**Fig. R31**) both indicate the changed chemical states of Ni and Co after the long-term seawater reduction tests, again confirming that the new active sites are still Ni and Co species.

Fig. R28 *In situ* Raman data of NCP/PC under reaction conditions in natural seawater. (a) Potential-dependent Raman spectra. (b) Time-dependent Raman spectra at a fixed applied potential.

Fig. R29 XPS survey scan analysis based on data before and after the natural seawater electrolysis. (a) Full range spectra. (b–d) Evidence of Na/K species from seawater left on the surface of the electrode after the test.

Fig. R30 High-resolution XPS spectra of NCP/PC after natural seawater reduction tests in the (a) Ni $2p_{3/2}$, (b) Co $2p_{3/2}$, (c) P $2p$, and (d) O $1s$ regions.

Fig. R31 Infrared signals of NCP/PC in the range of 500–4000 cm^{-1} before and after the natural seawater reduction test. The most obvious change occurs at wavenumbers lower than 1425 cm^{-1} (bands below 800 cm^{-1} may correspond to metal–oxygen

stretching and bending modes, please refer to: *J. Mater. Chem. A* **1**, 9046–9053 (2013)), as part of the phosphides convert to NiCo-based oxidized species (please refer to: *J. Colloid Interface Sci.* 608, 70–78 (2022), *Chinese Chem. Lett.* 33, 2741–2746 (2022), *Ceram. Int.* 40, 5339–5342 (2014), *Energy Fuels* **35**, 18815–18823 (2021), *Adv. Funct. Mater.* **17**, 644–650 (2007), *Sci. Rep.* **6**, 18737 (2016), *Int. J. Hydrogen Energy* **36**, 10057–10064, (2011)).

Comment 5: The diffusion of feedstocks/products from micro/mesopores to the bulk electrolyte should be quantitatively analyzed.

Response 5: Indeed, quantitatively analyzing such diffusions is an excellent suggestion, but quantifying the diffusion of feedstocks/products from micro/mesopores to the bulk electrolyte is a very difficult thing to do experimentally as well as theoretically. Current technologies do not allow for accurate quantification of seawater (as a feedstock) from the micropores/mesopores of the electrode to the bulk electrolyte.

Despite the difficulties, we have tried our best to answer your question by performing theoretical simulations. We have performed finite element simulations involving the models of the optimal sample (i.e., representing NCP/PC) and a counterpart electrode to try to quantitatively analyze the diffusion of feedstocks/products from pores to the electrolyte. Detailed methods of theoretical simulating were listed in **Supplementary Notes 5 and 6** and **Supplementary Figs. 55 and 56**. The solution is obtained according to the equations and boundary conditions shown in **Supplementary Note 6**, which include the velocity field (u), the pressure field (p), the gas phase volume fraction field (ϕ_g), and the precipitation concentration field (c) at each time step. Moreover, the gas phase volume fraction and velocity field quantitatively provide information about the motions of the bubbles. Where the gas phase volume fraction is close to 1, the fluid velocity can be regarded as the velocity of the gas (**Supplementary Fig. 58**). In addition, we defined the repelling ratio to precipitates as $\eta(t)$, which can be acquired via the following calculation equation:

$$\eta(t) = 1 - \frac{1}{c_0 V} \left(\int_{\Omega} c(t) dV + A \right)$$

Ω refers to the portion of the computational domain within the electrode, and A is the amount of precipitated material from time 0 to time t . Comparison of the repelling ratio requires a given characteristic time τ , its scale is equivalent to the period of seawater injection (i.e., both are equal in size by orders of magnitude), and empirically preferable $\tau = 0.001\text{s}$ to 0.1s . Repelling ratios at time $t = \tau$ quantitatively reflects the abilities of the electrodes to repel precipitates.

Supplementary Fig. 58 Comparison of the volume fraction of gas phase, the ϕ_g , for (a) the optimal electrode and (b) the counterpart electrode.

Notably, **Supplementary Fig. 59** shows the distribution results of precipitation concentration during the reaction. Due to a more directional movement of bubble flows in the optimal sample, the precipitates can be discharged efficiently (**Supplementary Fig. 59a**); however, in the counterpart sample, due to the non-directional flow of the electrolyte solution (i.e., more random directions, see **Supplementary Fig. 59b**), the precipitate discharge ability is not as good as that of the optimal sample. Furthermore, **Supplementary Fig. 60** shows that vortices are easy to appear in the velocity field, this consists with the lower precipitate discharge ability stemmed from non-directional flows in the counterpart sample. Additionally, we quantified the amount of precipitates

diffusing out of the pores for both two models and compared theoretical repelling ratios at different reaction times. **Supplementary Fig. 61** shows the repelling ratios of the two electrodes, with lower $\eta(t)$ for the counterpart sample.

Supplementary Fig. 59 Visualized comparison of distribution of the precipitation concentration. (a) Precipitation concentration distribution of the optimal electrode that represents NCP/PC and (b) the counterpart electrode.

Supplementary Fig. 60 Vortices that are prone to appear in the velocity field. The non-

directional flow field is not conducive to sediment repulsion.

Supplementary Fig. 61 (a) Quantitative comparison of the optimal electrode and the counterpart electrode in terms of the $\eta(t)$. (b) The relationship between the η and the gas production volume per unit area.

Comment 6: *The mechanical and electrical conductivity of 3D micro/nano-structured electrodes should be further checked. The optimization of pore size, pore thickness, and areal loading should be explained.*

Response 6: As suggested, we have provided more data regarding the mechanical properties and electrical conductivity of the NCP/PC. The mechanical properties of NCP/PC and PC were first evaluated using three-point bending tests (please see **Fig. R32**). Both samples display evidence of inelastic deformation, with strains at failure in the 1.37-to-1.6% range. The maximum flexural strength values of the two samples (11.4 MPa vs 13.8 MPa) are not much different. Besides, a piece of NCP/PC can bear the strain of a 200-g weight pulling on it (**Fig. R33**). Furthermore, the NCP/PC exhibits a higher conductivity than the bare PC ($0.03153 \text{ KS cm}^{-1}$ vs. $0.0019 \text{ KS cm}^{-1}$), according to the results of the four-probe conductivity test (**Fig. R34**).

Fig. R32 (a) Flexural stress–flexural strain curves for NCP/PC and PC. Inset illustrates the three-point bending test setup. (b) Flexural strength and modulus values of NCP/PC and PC.

Fig. R33 Photo of a piece of NCP/PC easily withstanding 200 grams of weight. This NCP/PC has a size of $\sim 20 \text{ mm} \times \sim 5 \text{ mm} \times \sim 1.9 \text{ mm}$.

Fig. R34 Four-probe conductivity measurement results for NCP/PC and PC. Error bars denote the standard deviation of experimental replicates.

We agree with you that the pore structure and loading should be further optimized, however, please be aware that the well-arranged, low-tortuosity micro-channels of the PC-based electrodes are naturally inherited from pine wood. Also, as a biomass-derived carbon substrate, pores of PC cannot be changed at will. In addition, we carefully and comprehensively compared different NCP-based electrodes: NCP/PC as the optimal sample has achieved the best performance for natural seawater reduction to date. In other words, compared to other NCP-based electrodes and perilously reported natural seawater reduction electrodes, NCP/PC with a record-high precipitation-repelling capacity already represents the most effective pore size, pore thickness, and areal loading. Most importantly, the primary focus of this work is to report on the natural biomass-derived honeycomb architectures that easily achieve precipitation-less natural seawater reduction. The further and thorough optimization of pore size, pore thickness, and areal loading will be done in our future work.

Although it is difficult to regulate the pore of PC experimentally, we have performed finite-element method simulations in an attempt to obtain theoretical results for pore

structure optimization to further improve the quality of our work. The corresponding results are given in **Figs. R35–38** as well as **Supplementary Figs. 62–65**. We defined the repelling ratio to precipitates as $\eta(t)$. Using calculation methods in **Supplementary Notes 5 and 6**, the anti-precipitation abilities of different samples with a series of pore widths are measured. At shorter reaction times, smaller pores are reasonably more advantageous/dominant in repelling precipitates because they are filled bubbles faster (**Fig. R35a**). The optimum pore width (pore size) increases with the increase of reaction time, and the optimum pore size was near 10 μm (**Fig. R35b**). In addition, we have calculated and compared the gas phase distribution in pores of different widths (**Fig. R36**), which is consistent with **Fig. R35**. Furthermore, at a fixed pore width of 10 μm , we considered and compared abilities of samples with different pore depths to repel the precipitates. Theoretically, the shallower the pore, the easier it is to discharge the precipitate (**Fig. R37a,b**). However, the reaction area (i.e., reaction site numbers) of the electrode is proportional to the pore depth. Notably, if the overall performance of the electrode is indirectly obtained/described by multiplying the repelling ratio and the depth of the pore, the performance index increases with the increase of the pore depth (**Fig. R37c**). As shown in **Fig. R38**, the gas phase distribution of electrode samples with different pore depths is visually compared.

Fig. R35 Comparison of η for different pore widths. (b) Determination of optimal pore width.

Fig. R36 Direct comparison and quantification of gas phase distribution in pores of different widths.

Fig. R37 (a) Relationship between repelling ratio and time at different pore depths. (b) Relationship between repelling ratio and pore depth at different times. (c) Comparison of the performance at different times by multiplying the repelling ratio and the depth of the pore.

Fig. R38 Direct comparison and quantification of gas phase distribution in pores of different depths.

Comment 7: How to anchor taps on the structured electrocatalysts? What is the actual current density distribution and related gas evolution?

Response 7: It is not necessary to anchor taps on structural electrocatalysts. Regarding the actual current density distribution and related gas evolution, that is a very thoughtful question. The actual current density distribution of a 3D electrode with precipitation generation during natural seawater electrolysis can hardly be known with the present technical means. However, the evolution of gas bubbles on the electrode surface can be observed by using *operando* microscope observation (please see **supplementary video 1**). Since the current should not exist where the bubbles are attached to the electrode surface (i.e., water molecules cannot accept any electrons from the electrocatalyst), the distribution of the bubbles on the electrode surface can actually reflect the current density distribution to a certain extent. That is to say, the real current density distribution is never static, and your question raises some excellent points.

On the other hand, to improve the quality of our work and to more fully address your concerns, we referred to previous studies that used infrared thermography to map the current density distribution of electrodes. We performed this measurement in a three-electrode system. The heat generated on the catalyst surface by the charge transfer will be captured in time and presented in the form of a heat map. The quantity of heat (q) generated on the catalyst surface as a result of charge transfer is linearly correlated with the current density ($q \propto j$). For an individual electrode region, the occurrence of localized activity should result in proportional local heat generation. Thus, the observation of the localized heating distribution can in principle be regarded as an electrochemical activity distribution (please refer to: *ACS Energy Lett.* **7**, 2410–2419 (2022), *Nat. Commun.* **14**, 6579 (2023)). The heat map is directly related to the current distribution. At the eNSR reduction j of 200 mA cm^{-2} , the relatively uniform heat map is equivalent to a relatively uniform current density distribution on the surface of NCP/PC (the 5-min sample or 60-min sample in **Fig. R39b** or **Supplementary Fig. 66b**).

Fig. R39 Mapping eNSR activities (current densities) on our NCP/PC electrode (size: $\sim 2 \text{ cm} \times \sim 3 \text{ cm}$) using infrared thermography. (a) Mapping eNSR activities under different current densities as well as (b) different reaction times. Since the experiments were conducted in winter, the temperatures were lower.

Comment 8: *What is the maximum current density on the electrocatalysts in an industrial device?*

Response 8: Please see the Response 1.

REVIEWERS' COMMENTS

Reviewer #1 (Remarks to the Author):

The current version can be published.

Reviewer #2 (Remarks to the Author):

The response is very nice. The revised manuscript can be accepted as it is.

Reviewer #3 (Remarks to the Author):

After checking the revised main text and response letter, I still have many concerns on the novelty and impact of this work.

Firstly, it is very easy to achieve low overpotentials and large areal current density. It should be noticed the electrocatalysis performance is very sensitive to the size and many considerations for seawater reduction electrocatalysis at industrial-level is still lacking.

Secondly, the analysis on the actual seawater is not clear. Also how to define the ultrastable electrocatalysis is an open question.

Thirdly, the origin of seawater reduction (SR) reactivity on NiCoP decorated PC is unclear yet. The evolution of local environmental electrolyte and ion/electronic conductivity should be carefully considered at industrial-level.

Fourthly, how to maintain the H₂ bubble with limited size and maintain good contacts between the active NiCoP and electrolyte should be well explained in an industrial reactor. The current H₂ generation meets many challenges for industrial reactors, but outstanding data always presented in small electrolyzes at industrial-level current densities.

Reviewer #1 (Remarks to the Author): *The current version can be published.*

Response: We sincerely appreciate the positive comments and the valuable feedback provided by the reviewer #1.

Reviewer #2 (Remarks to the Author): *The response is very nice. The revised manuscript can be accepted as it is.*

Response: We sincerely appreciate the professional suggestions from the reviewer #2 and the kind recommendation of our research.

Reviewer #3 (Remarks to the Author): *After checking the revised main text and response letter, I still have many concerns on the novelty and impact of this work.*

Firstly, it is very easy to achieve low overpotentials and large areal current density. It should be noticed the electrocatalysis performance is very sensitive to the size and many considerations for seawater reduction electrocatalysis at industrial-level is still lacking.

Secondly, the analysis on the actual seawater is not clear. Also how to define the ultrastable electrocatalysis is an open question.

Thirdly, the origin of seawater reduction (SR) reactivity on NiCoP decorated PC is unclear yet. The evolution of local environmental electrolyte and ion/electrical conductivity should be carefully considered at industrial-level.

Fourthly, how to maintain the H₂ bubble with limited size and maintain good contacts between the active NiCoP and electrolyte should be well explained in an industrial reactor. The current H₂ generation meets many challenges for industrial reactors, but outstanding data always presented in small electrolyzes at industrial-level current densities.

Response: The main focus of our research is on the design of genuine anti-precipitation seawater reduction cathodes, and we give thorough discussions and strong experimental support for underlying reasons why the MBPTS-based electrode rejects precipitation proficiently. The NCP/PC we report in this work will be the most powerful electrode available for repelling Ca²⁺/Mg²⁺ precipitates.

Firstly, using real natural seawater with limited H⁺/OH⁻ as the electrolyte would certainly make it difficult to produce low overpotentials and high areal current densities (please refer to *Energy Environ. Sci.* **10**, 788–798 (2017), *Energy Environ. Sci.* **11**, 1898–1910 (2018), *Nat. Energy* **8**, 264–272 (2023)). Most importantly, the emphasis of the work is on achieving anti-precipitation reduction of natural seawater, which was truly difficult to achieve in previous studies (please refer to one of best studies so far: *Nat. Energy* **8**, 264–272 (2023)). High-speed camera videos, in situ Raman data, ex situ characterization data, and theoretical simulations all confirm the strong precipitation resistance of our electrode and the effectiveness of our strategy. Besides, please note that we have presented polarization curves of NCP/PC in natural seawater reduction with different sizes, and the small difference in current densities confirms the successful scale-up of electrode synthesis.

Secondly, our focus is on repelling precipitates, and we have provided the source, pH value, as well as Ca and Mg contents in the seawater. Therefore, the analysis on actual

seawater is certainly clearer than most previous studies on natural seawater reduction. Please note that the ultrastable eNSR cathode is not an unjustified claim, but holds true relative to previously reported natural seawater reduction cathodes. Among the known cathodes for electrochemical natural seawater reduction, our NCP/PC cathode does demonstrate the strongest ability to repel the $\text{Ca}^{2+}/\text{Mg}^{2+}$ precipitates and, therefore, better maintain the exposure of active sites. Again, we reiterate that we used completely untreated natural seawater. We used the natural seawater directly for eNSR electrolysis demonstration without filtering it. Note that previous natural seawater electrolysis work (please refer to: *Nat. Energy* **8**, 264–272 (2023)) used filtered natural seawater, while this work did not filter natural seawater, allowing for genuine direct natural seawater reduction. In terms of anti-precipitation ability, it is safe to say that this electrode is among the best ones in the field of natural seawater reduction.

Thirdly, NCP itself is known as a classical catalyst for hydrogen evolution reaction, and its reactivity origin has been extensively reported in the literature (please refer to *Nano Lett.* **16**, 7718–7725 (2016), *Adv. Funct. Mater.* **26**, 6785–6796 (2016), *Adv. Mater.* **29**, 1605502 (2017), *Adv. Energy Mater.* **9**, 1901213 (2019), *Appl. Catal. B* **343**, 123579 (2024)). Moreover, the most important implication of this work is not to investigate the reactivity origin of NCP. Please note that the evolution of local environmental electrolyte at the surface of seawater reduction electrodes when operated at industrial-level current densities has been extensively described in previous literature (for instance: *ChemSusChem* **9**, 962–972 (2016), *Nat. Energy* **5**, 367–377 (2020), *Nat. Energy* **8**, 264–272 (2023)). While carbonates in seawater can function as buffers, their capacity is insufficient to stop local pH increases at the cathode. Studies revealed that pH changes near the electrode surface could be as much as 5–9 pH units different from the bulk seawater when the medium's overall pH value is between 4 and 10, even at moderate current densities $<10 \text{ mA cm}^{-2}$ (please refer to *Nat. Energy* **5**, 367–377 (2020)). Under industrial-level electrolysis conditions, protons can be produced in large quantities on the anode surface and hydroxide ions can be produced in large quantities on the cathode surface (this can also lead to increased ion conductivity). Due to an increase in local acidity, we used a commercial dimensionally stable anode for seawater oxidation and surface precipitation-resistant NCP/PC as the cathode, which ultimately resulted in a good overall natural seawater splitting performance. In addition, the catalytic surfaces will certainly be reconstructed during industrial-level testing (please see Supplementary Note 9), bringing about a change in electrical conductivity, but this factor is not a critical one in influencing the anti-precipitation ability of NCP/PC. Importantly, our catalysts did not show significant performance degradation even when tested in the electrolysis stacks with a total electrode area of 54 cm^2 .

Fourthly, just the relatively good overlap of ECSA-normalized polarization curves before and after the eNSR test (Supplementary Fig. 31) is enough to confirm that the activity of NCP/PC is not severely decreased for at least 10-h operation under the large current density of 500 mA cm^{-2} . Besides, the uniform distribution of reduction current on the NCP/PC surface (in situ infrared imaging data) and the trace precipitation (in situ Raman data) further suggest that our NCP/PC already maintains the released H_2 bubbles with limited sizes and the good electrolyte-NCP catalyst contact under working

conditions. In fact, the electrode area for our assembled electrolysis stacks (54 cm²) has far exceeded the best previous report (9 cm², *Nat. Energy* **8**, 264–272 (2023)). It is important to note that it was not easy to realize this scale of testing with natural seawater because it involves various issues such as scale-up of electrode synthesis, stopping fluid leakage under high current density operation, and preventing rejected precipitates from clogging the flow tubes. Therefore, we customized our reactor with a lot of design ideas. For instance, in order to minimize the hydraulic resistance and enhance the flow rate of the liquid, not only did we not use the traditional serpentine flow paths for the bipolar plates, but we adopted a modified straight flow path with a larger groove width and groove depth. Besides, we also assembled gaskets with a series of thicknesses and tried different torque (e.g., 4 Nm, 5 Nm, 6 Nm) to achieve a more stable electrolysis process with our stacks, i.e., the catalyst can better maintain a stable bubble release and good contact with the electrolyte. In addition, we equipped the electrolyzer stack with tubes having a large enough inner diameter to prevent the precipitate repelled by NCP/PC from clogging in the flow path/tube. The results achieved by our work are already a big step forward compared to current research work related to natural seawater reduction, and the direct industrial demonstration of several years of operation will be one of the most vital ultimate goals in our future endeavors. This is because it requires ongoing improvement and testing of electrolysis stack structures for the reaction (for instance, we will upgrade the titanium alloy plates currently used to make electrolysis stacks more suitable for durability testing), which is not the real focus of this work.